



# European heatwaves in present and future climate simulations: A Lagrangian analysis

Lisa Schielicke[1,2] and Stephan Pfahl[1]

[1]Institut für Meteorologie, Freie Universität Berlin, Berlin, Germany
[2]Institut für Geowissenschaften, Abteilung Meteorologie, Universität Bonn, Bonn, Germany

**Correspondence:** Lisa Schielicke (lisa.schielicke@met.fu-berlin.de)

**Abstract.** Heatwaves are prolonged periods of anomalously high temperatures that can have devastating impacts on the environment, society and economy. In the recent history, heatwaves have become more intense and more numerous over most continental areas and it is expected that this trend will continue due to the ongoing global temperature rise. This general intensification may be modified by changes also in the underlying thermodynamical and dynamical processes. In order to study

potential changes in heatwave characteristics and dynamics, we compare Lagrangian backward trajectories of air streams associated with historic (1991-2000) and future (2091-2100) heatwaves in six different European regions. We use a percentile-based method (Heat Wave Magnitude Index daily) to identify heatwaves in a large ensemble of climate simulations (Community Earth System Model - Large Ensemble (CESM-LE) with 35 members). In general, we find that air parcels associated with heatwaves are located to the east or inside the respective regions three days prior to the events. For future heatwaves, the model projects a

north-/northeastward shift of the origin of the air masses in most study regions. Compared to climatological values, air streams associated with heatwaves show a larger temperature increase along their trajectory, which is connected to stronger descent and/or stronger diabatic heating when the air parcels enter the boundary layer. We find stronger descent associated with adiabatic warming in the northern, more continental regions and increased diabatic heating in all regions (except of the British Isles) in the simulated future climate. The enhanced diabatic heating is even more pronounced for heatwaves over continental

regions. Diabatic temperature changes of near-surface air are driven by sensible heat fluxes, which are stronger over dry soils. The amplified diabatic heating associated with future heatwaves may thus be explained by an additional drying of the land surface.

## 1 Introduction

Heatwaves can have huge socio-economic impacts including an increased mortality (Guo et al., 2017), an enhanced risk of

wildfires due to hot and dry conditions accompanying heatwaves (Jones et al., 2020), crop failures in connection with droughts (Zampieri et al., 2017) and damage to the flora in general (Breshears et al., 2021). Impacts to the water, transport and energy infrastructures can lead to tremendous economic losses that are expected to increase in the future (see Forzieri et al., 2018, who studied these impacts for Europe). In their Sixth Assessment Report, the Intergovernmental Panel on Climate Change (IPCC) states that "hot extremes (including heatwaves) have become more frequent and more intense across most land regions since





25 the 1950s" (IPCC, 2021). Due to the expected ongoing rise in the global mean temperature in the 21st century, this progress is expected to continue (Meehl and Tebaldi, 2004). It is therefore of general interest to study the differences between historic and future heatwaves. Understanding changes in their dynamics, variability and underlying processes can further help to quantify future changes in heatwave occurrence and to take measures to protect against their impacts.

 In general, heatwaves are long periods of unusually high temperatures (Perkins and Alexander, 2013). However, there is no
30 common, global heatwave definition and thresholds differ from region to region due to locally different climates. Heatwave identification can be based on absolute or relative thresholds of a certain temperature measure, for example the daily maximum or minimum temperature (see, e.g. Perkins and Alexander, 2013; Perkins, 2015, for a discussion of different heatwave definitions). Absolute thresholds are fixed and therefore do not account for a general increase in mean temperatures. Hence, an increase in global mean temperature will naturally lead to a higher number of identified heatwave events if these thresholds are
35 not adjusted to the "new climate". Indeed, Vogel et al. (2020) observe an increase in heatwaves for fixed temperature thresholds, but they only see relatively small changes for moving, i.e. temporally varying, thresholds. In order to investigate future changes in heatwaves beyond mean climate warming, we will take a relative, percentile-based threshold (after Russo et al., 2015) that is adapted to the respective climate. This allows for the identification and comparison of heatwaves in different (European) regions, climates and time slices with different background temperatures.

40 Dynamically, heatwaves are linked to quasi-stationary atmospheric blocking patterns (e.g. Pfahl and Wernli, 2012, who showed this relationship for the Northern hemisphere) or extended ridges (Sousa et al., 2018). Such anticyclonic circulation anomalies lead to the build-up of warm temperature extremes in summer due to the associated subsidence and adiabatic warming, the clear-sky conditions that lead to surface heating by solar radiation, low wind speeds and warm-air advection (Meehl and Tebaldi, 2004; Pfahl, 2014; Bieli et al., 2015; Zschenderlein et al., 2019; Kautz et al., 2021). Moreover, soil moisture-
45 atmosphere feedbacks enhance the heating of lower tropospheric air in the planetary boundary layer due to the increase of sensible heat fluxes and decrease of latent heat fluxes associated with reduced soil moisture content during (prolonged) drought and heatwave conditions (Fischer et al., 2007; Seneviratne et al., 2010; Stéfanon et al., 2014; Miralles et al., 2019).

 Several previous studies have shown that detailed insights into the processes leading to hot extremes and heatwaves can be obtained from a Lagrangian description of the associated air mass trajectories (Bieli et al., 2015; Santos et al., 2015; Quinting
50 et al., 2018; Schumacher et al., 2019; Zschenderlein et al., 2019, 2020; Catalano et al., 2021). Such a Lagrangian approach allows for differentiating between thermodynamic and dynamic contributions to the warming of air masses. Investigating 10-day backward trajectories initiated close to the surface, Bieli et al. (2015) found that hot events are connected to strong adiabatic and diabatic warming, but only to weak northward transport of warm air from the south towards their European study regions (British Isles, central Europe and the Balkans region). Furthermore, Bieli et al. (2015) found that hot extremes in the
55 regions closer to the ocean (British Isles and Balkans) are often dominated by dynamical processes, i.e. adiabatic warming due to descent, while in their continental region (Central Europe) local, diabatic heating by surface fluxes is more important. Zschenderlein et al. (2019) extended the Lagrangian investigation to six different European regions and additionally clustered the trajectories with respect to the prevailing diabatic warming or cooling and strong or weak descend occurring along the air parcel trajectories. They confirmed the finding of weak horizontal transport associated with heatwaves. Schumacher et al.





(2019) used backward trajectories to show that heatwaves can be intensified due to enhanced sensible heat fluxes in upwind regions affected by drought conditions.

The above cited Lagrangian heatwave studies focused on the analysis of past events based on reanalysis data and did not investigate potential future changes in these Lagrangian dynamics. In principle, future changes in heatwaves beyond the mean climate warming may be due to changes in the associated atmospheric dynamics (e.g., varying transport patterns or adiabatic

warming) or thermodynamics (e.g., altered sensible heating near the surface). Schaller et al. (2018) and Brunner et al. (2018) showed that the link between atmospheric blocking and European heatwaves appears to be relatively stable in ensemble simulations of future climate, suggesting that dynamical changes might be of minor importance. Also Vogel et al. (2020) concluded that future changes in heatwaves are mainly driven by thermodynamic processes and changes in the underlying atmospheric dynamics are small. Rasmijn et al. (2018) hypothesize a disproportional intensification of future (continental) heatwaves in

western Russia due to amplified sensible heating linked to soil moisture feedbacks.

In this study, a detailed Lagrangian analysis of future changes in the processes associated with European heatwaves is presented. In particular, we investigate the following research questions:

- How will the atmospheric transport patterns associated with European heatwaves change in a warming climate?

- Will changes in thermodynamic or dynamic processes lead to an amplification of heatwaves beyond mean warming?

Since heatwaves are rare events, we use an ensemble data set of climate simulations with 35 members to increase the number of heatwave events in a historic (1991-2000) and a future (2091-2100) time slice. The data and methods including the definition of the regions and the identification of heatwaves are described in section 2. The results are presented and discussed in section 3 followed by a summarizing discussion and conclusion in section 4.

## 2 Data and Methods

In this section, we give an overview of the data used (section 2.1) and define the six European study regions (section 2.2). The heatwave identification follows the work of Russo et al. (2015) adapted to the ensemble data and is described in section 2.3. Finally, the trajectory calculations and clustering using the Lagrangian tool LAGRANTO (Sprenger and Wernli, 2015) is described in section 2.4.

### 2.1 CESM-LE data

In this study, we use data from model simulations based on the Community Earth System Model (CESM) - Large Ensemble (CESM-LE) project (Kay et al., 2015) with 35 ensemble members. The ensemble members have been generated through small differences in their initial conditions – mainly by adding random perturbations on the order of $10^{-14}$ K to the initial air temperature fields. The simulations have been externally forced by the historical (up to the year to 2005) and representative concentration pathway (RCP) 8.5 (years 2006 to 2100) conditions. The atmospheric variables of the CESM-LE data have a

horizontal grid spacing of approximately $\approx 1°$ in latitudinal and $1.25°$ in longitudinal direction and 30 hybrid vertical levels.



Re-runs of the simulations have been performed for two 10-year time slices, 1991–2000 (historic) and 2091–2100 (future), based on restart files from the original CESM-LE simulations and using the exact same model setup, in order to obtain additional output fields (in particular of six-hourly vertical wind on model levels) required for the trajectory calculations. The identification of the heatwaves is based on the temperature at reference height (2 meters above ground). Furthermore, we use

six-hourly three-dimensional wind fields as well as pressure, potential temperature, and air temperature to study heatwave properties and calculate parcel trajectories with LAGRANTO (see section 2.4 for more details).

## 2.2 Definition of study regions

**Table 1.** Definition of regions representing different European climates (following Zschenderlein et al., 2019, with adaptions to the CESM-LE data). Latitudes are rounded.

| Name | Full name | Longitudes (°E) ($\Delta$lon $= 1.25$°E) | Latitudes (°N) ($\Delta$lat $\approx 0.94$°N) | Total number of grid-points ($\Delta x \times \Delta y$) | Number of grid-points over land |
|------|-----------|-------------|------------|---------------------|----------------|
| BI | British Isles | −10.0 to 2.5 | 48.5 to 58.9 | 132 $(11 \times 12)$ | 76 |
| CE | Central Europe | 3.75 to 16.25 | 44.8 to 55.1 | 132 $(11 \times 12)$ | 121 |
| GI | Greece and Italy | 10.0 to 25.0 | 36.3 to 44.8 | 130 $(13 \times 10)$ | 91 |
| IP | Iberian Peninsula | −10.0 to 2.5 | 36.3 to 44.8 | 130 $(11 \times 10)$ | 85 |
| Sc | Scandinavia | 5.0 to 20.0 | 57.0 to 65.5 | 130 $(13 \times 10)$ | 104 |
| WR | Western Russia | 33.75 to 46.25 | 47.6 to 58.0 | 132 $(11 \times 12)$ | 132 |

We study heatwaves in historic and future time slices in six European regions with different climates: the British Isles (BI), Central Europe (CE), Greece and Italy (GI), the Iberian Peninsula (IP), Scandinavia (Sc) and Western Russia (WR). Thereby,

we follow the definitions of the regions by Zschenderlein et al. (2019). Details on the six regions with respect to the CESM-LE data are summarized in Table 1. The study regions are marked in Figure 1.

## 2.3 Identification of heatwaves

A *heatwave* is defined, if a specific temperature threshold at a grid point is exceeded for at least 3 consecutive days in a large area. In our analysis, this temperature threshold is given by the 90th percentile of the daily maximum temperature (calculated

as the maximum over four six-hourly time steps) within a 30 day window around the day of interest. The 90th percentile is derived by taking into account all years in the historic or in the future time slice simulations, respectively. For example, for the calculation of the daily maximum temperature percentiles of 15 August, the 30 day period around the date, i.e. the period of 1 to 31 August, in all 350 years (35 members times 10 years per time slice) are considered (see Fig. 1). Accordingly, heatwaves are defined with respect to the historic climate (percentiles calculated form the historic period) in the 1991–2000 time slice and

with respect to future climate in the 2091-2100 time slice.

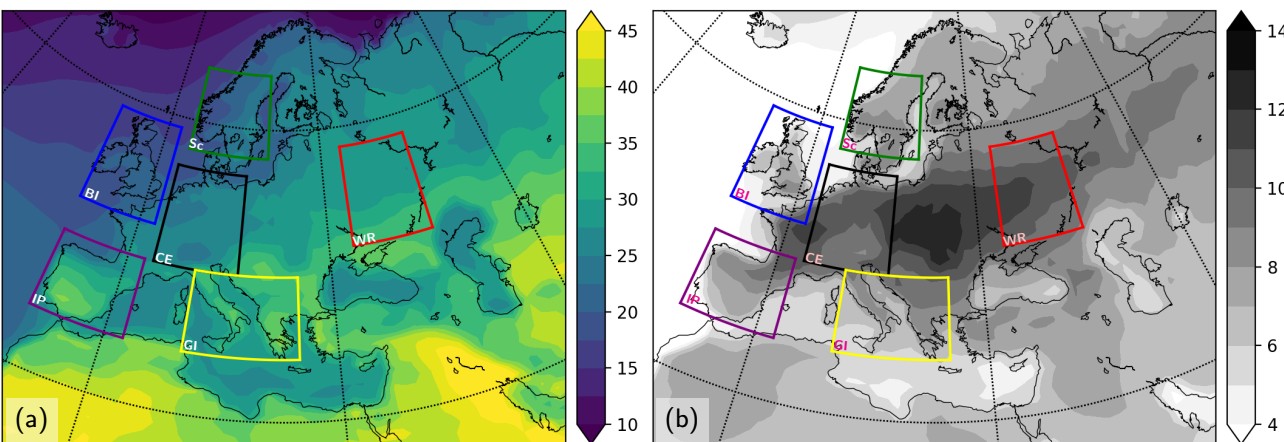

**Figure 1.** 90th percentile of daily maximum 2m-temperature in August: (a) historic time slice (1991-2000), in degrees Celsius; (b) difference (in K) between future (2091-2100) and historic time slices. Shown is the 90th percentile centered on 15 August, which is derived by taking into account all daily maximum temperatures from 1-31 August (15 August $\pm 15$ days) in the respective time slice. Colored boxes show the outline of the different European regions (see Table 1)

.

In order to estimate the magnitude of a heatwave, the percentile-based Heat Wave Magnitude Index daily (HWMId) is used, as defined by Russo et al. (2015). It is based on the calculation of the *Daily Heat Wave Magnitude* $M_d$, which is given as

$$
M_d(T_{d,max}) = \begin{cases} \dfrac{T_{d,max} - T_{\Delta y,25th}}{T_{\Delta y,75th} - T_{\Delta y,25th}} & : \text{if } T_{d,max} > T_{\Delta y,25th} \\ 0 & : \text{otherwise} \end{cases} \tag{1}
$$

with daily maximum temperature $T_{d,max}$ and the $25th$ ($75th$) percentile of annual maximum temperature $T_{\Delta t,25th}$ ($T_{\Delta y,75th}$)

that occurred within a time range of $\Delta y$ years (e.g. $\Delta y = 30$ years in Russo et al., 2015). In our case, the time range $\Delta y$ is given by each 10 year time slice considering all 35 ensemble members, i.e. $\Delta y = 350$ years for the historic and the future run, respectively. The $M_d$ values are calculated at each grid point, and summing them up gives the HWMId per day in a specific region. In order to account for the large-scale character of heatwaves, we additionally use a minimum area criterion of at least 5 % of all grid points over land that must satisfy $M_d > 0$. The latter criterion ($M_d > 0$) also makes sure that heatwaves occur

during the warmest time of the year, typically the summer months. Note that we detect more heatwave days towards the end of the two time slices due to the general temperature increase during the 10-year periods.





## 2.4 Trajectory calculation and clustering

LAGRANTO (Sprenger and Wernli, 2015) is a Lagrangian analysis tool that allows the calculation of backward and forward trajectories based on various data sets. LAGRANTO can be used to interpolate other variables of interest such as temperature

or potential temperature along the trajectory paths and thus study physical processes, such as diabatic and adiabatic heating, in three-dimensional air streams. In this work, we use LAGRANTO to calculate 10-day backward trajectories. These air parcels are started 10, 30, 50, and 100 hPa above the surface at land grid points that satisfy the heatwave criterion. The trajectories are started at 12 UTC at each heatwave day in the respective region. In order to compare the heatwave trajectories with typical climatological values, we additionally start 10-day backward trajectories at 12 UTC on each day in the boreal summer period

(JJA: June, July, August) at 10, 30, 50 and 100 hPa above ground. The output of various variables along the trajectories as well as the air parcel locations are saved every 6 hours, with the time $t = 0$ referring to the initialization of the trajectory at the heatwave location. In order to classify the air parcel trajectories, the following properties are determined along the trajectories:

- Maximum change in potential temperature $\Delta\theta_{max} = \max(\text{abs}[\theta_0 - \theta(t)]) \cdot \text{sign}[\theta_0 - \theta(t)]$ with respect to the starting value $\theta_0 := \theta(t = 0)$. Positive (negative) values indicate warming (cooling) along the trajectory,

- Maximum change in temperature $\Delta T_{max} = \max(\text{abs}[T_0 - T(t)]) \cdot \text{sign}[T_0 - T(t)]$ with respect to the starting value $T_0 := T(t = 0)$,

- Change in pressure $\Delta P_{3d} = P(t = 0h) - P(t = -72h)$ in the first 3 days after initialization of the backward trajectory.

Note that the maximum changes can be positive as well as negative. The first two criteria are similar to Zschenderlein et al. (2019). The pressure criterion is adapted here to more directly reflect the vertical motion in the period before the arrival at the

heatwave location.

**Table 2.** Criteria for trajectory clusters following Zschenderlein et al. (2019), but with a different pressure criterion.

| Cluster | $\Delta T_{max}$ | $\Delta\theta_{max}$ | $\Delta P_{3d}$ |
|---------|------------------|----------------------|-----------------|
| A | $> 0$ | $< 0$ | not considered |
| Bwd | $> 0$ | $> 0$ | $< 50\,\text{hPa}$ |
| Bsd | $> 0$ | $> 0$ | $\geq 50\,\text{hPa}$ |
| C | $< 0$ | $< 0$ | not considered |
| D | $< 0$ | $> 0$ | not considered |

The trajectories are clustered with respect to their thermodynamic ($\Delta T_{max}, \Delta\theta_{max}$) and dynamic ($\Delta P_{3d}$) properties. Individual changes in temperature $DT/Dt$ along a trajectory can be caused by diabatic or adiabatic processes. This becomes obvious by looking at the temperature tendency equation that follows from the first law of thermodynamics. In pressure-coordinates





$(x, y, p)$, the equation reads:

$$
\underbrace{\frac{DT}{Dt}}_{\substack{individual \\ tendency}} = \underbrace{\frac{\partial T}{\partial t}}_{\substack{local \\ tendency}} + \underbrace{\mathbf{v} \cdot \nabla T}_{advection} = \underbrace{-\frac{\alpha}{c_p}\omega}_{\substack{adiabatic \\ compression \\ or\ expansion}} + \underbrace{\frac{\dot{q}}{c_p}}_{\substack{diabatic \\ heating \\ or\ cooling}} \tag{2}
$$


where $\mathbf{v} = (u, v, \omega)^T$ is the three-dimensional wind vector with vertical wind component $\omega = Dp/Dt$. The nabla operator in pressure coordinates is given as $\nabla = (\partial/\partial x, \partial/\partial y, \partial/\partial p)^T$. Furthermore, $c_p$ is the specific heat capacity at constant pressure, $\alpha$ is the reciprocal of the density and $\dot{q}$ is the specific heating rate. Potential temperature is defined as

$$
\theta = T \left( \frac{p_0}{p} \right)^{R_l/c_p} \tag{3}
$$

with $p_0 = 1000$ hPa and specific gas constant of dry air $R_l$. The potential temperature tendency equation then follows from Eq. (2) using definition (3) as

$$
\underbrace{\frac{D\theta}{Dt}}_{\substack{individual \\ tendency}} = \underbrace{\frac{\partial \theta}{\partial t}}_{\substack{local \\ tendency}} + \underbrace{\mathbf{v} \cdot \nabla \theta}_{advection} = \underbrace{\frac{\theta}{T}\frac{\dot{q}}{c_p}}_{\substack{diabatic \\ heating \\ or\ cooling}} \tag{4}
$$

Hence, potential temperature changes along the trajectory path can only be caused by diabatic processes such as radiative heating or cooling or latent heat release during phase changes of water. Temperature changes along the trajectories can additionally

occur due to adiabatic compression or expansion (right-hand side of Eq. (2)) caused by vertical descent or ascent, respectively. The classification of the trajectories considers these aspects through the following definition of clusters (see Table 2 for an overview): Cluster A has a positive temperature difference, i.e. an increase in temperature along the trajectory, but a negative potential temperature difference, that could be caused, e.g., by radiative cooling. Cluster B has positive temperature and potential temperature (diabatic heating) differences. This cluster B is split further in two subclusters that are distinguished by the

maximum descent (or ascent) $\Delta P_{3d}$ in the three days prior to the heatwave day. If this value exceeds or equals 50 hPa, then the trajectory is in the Bsd cluster where "sd" stands for strong descent. Otherwise, the trajectory is classified as Bwd with "wd" indicating weak descent or even ascent if $\Delta P_{3d} < 0$. In cluster C both temperature and potential temperature changes along the trajectory are negative. Cluster D is composed of trajectories with negative temperature difference and positive potential temperature difference (diabatic heating) along the trajectory. Note that the vast majority of heatwave trajectories is associated

with clusters A, Bwd and Bsd, and the remaining clusters C and D are thus not considered in detail in the remainder of this study.

Finally, it should be noted that, although adiabatic and diabatic processes determine changes in temperature and potential temperature along a trajectory, for the local temperature change at the ground advection plays an important role, too (see again Eq. (2)). This means that air parcels for which the temperature does not change substantially during transport can still lead to

high temperatures at the target location if they originate in a warmer region.





## 3   Results

We first provide a general overview of the temperature increase in Europe expected under RCP 8.5 conditions in section 3.1. Afterwards we compare general heatwave properties in historic and future simulations. A first impression of changes in the associated dynamics is given by composite plots of the 500 hPa geopotential height level during heatwave onsets (section 3.2).
We then analyse the origins of the different air parcel clusters associated with historic and projected future heatwaves, also in comparison to the climatology (section 3.3). Finally, we take a closer look at future changes of dynamic and thermodynamic properties of the Lagrangian backward trajectories (section 3.4).

### 3.1   Future temperature increase in Europe in summer

The 90th percentile of daily maximum temperature for mid-August (Fig. 1(a)) reveals a latitudinal dependence with higher
values towards the south and generally higher values over land. Towards the end of the 21st century (Fig. 1(b)), the largest increases in this percentile ($> 10\,\mathrm{K}$) occur over the central parts of Europe, while the increase is smaller over southern and northern Europe (mainly $< 9\,\mathrm{K}$) and the British Isles (about $6\,\mathrm{K}$). The smallest warming is projected over the ocean ($< 6\,\mathrm{K}$), with a slightly higher temperature increase over the Mediterranean Sea compared to the North Atlantic. The average increase in the mean summer (JJA) temperature over land in the Euro-Atlantic region is reported to be between about 3-6 K (Chan et al.,
2020), indicating that increase of the 90th percentile is amplified compared to this mean increase over most regions.

Mean temperatures are calculated in the respective regions on detected heatwave days and compared to the climatological mean JJA temperatures in Fig. 2. The mean JJA temperatures are projected to increase in the future from about 4 K in BI (Fig. 2d) to about 8 K in WR (Fig. 2c). In BI, GI and IP (Fig, 2d-f), i.e. the regions that also cover large areas over the ocean, the temperature anomalies on heatwave days with respect to the mean JJA temperature for the future and historic time slices are of
a similar magnitude as the future increase of the mean JJA temperature. This also indicates that the projected warming during heatwaves is similar to the average JJA temperature increase. In contrast, in the regions that cover more grid points over land, i.e. Sc, CE and WR (Fig. 2a-c), the temperature anomalies during heatwaves are higher than the future increase of mean JJA temperatures and the anomalies increase even further in the future by about 1 K on average in Sc (Fig. 2a) and by about 2 K on average in CE and WR (Fig. 2b,c).
In general, we observe no or only minor differences between general heatwave properties in historic and future simulations (compare orange and blue boxes in Fig. 3). Likewise to Zschenderlein et al. (2019), we find a majority of zero to two heatwave occurrences per year in most regions (Fig. 3(a)). The typical duration is generally less than 10 days, but can reach maxima of up to about 20 days in IP in the historic time slice and about 45 days in GI and Sc in the future (Fig. 3(c)). In most regions, except for WR, the maximum duration increases in the future. Compared to Zschenderlein et al. (2019), the interquartile range
of heatwave days per year is smaller, especially in WR (ca. 12 vs. 20) and CE (ca. 8 vs. 14) where the first number refers to the data presented in Fig. 3(c) and the second number is the approximate value from Zschenderlein et al. (2019) (their Fig. 4).

Since the method of heatwave detection is percentile-based, the total number of heatwave days is – as to be expected – almost identical in both time slices. However, in the future period we observe an increase in the number of heatwave days towards



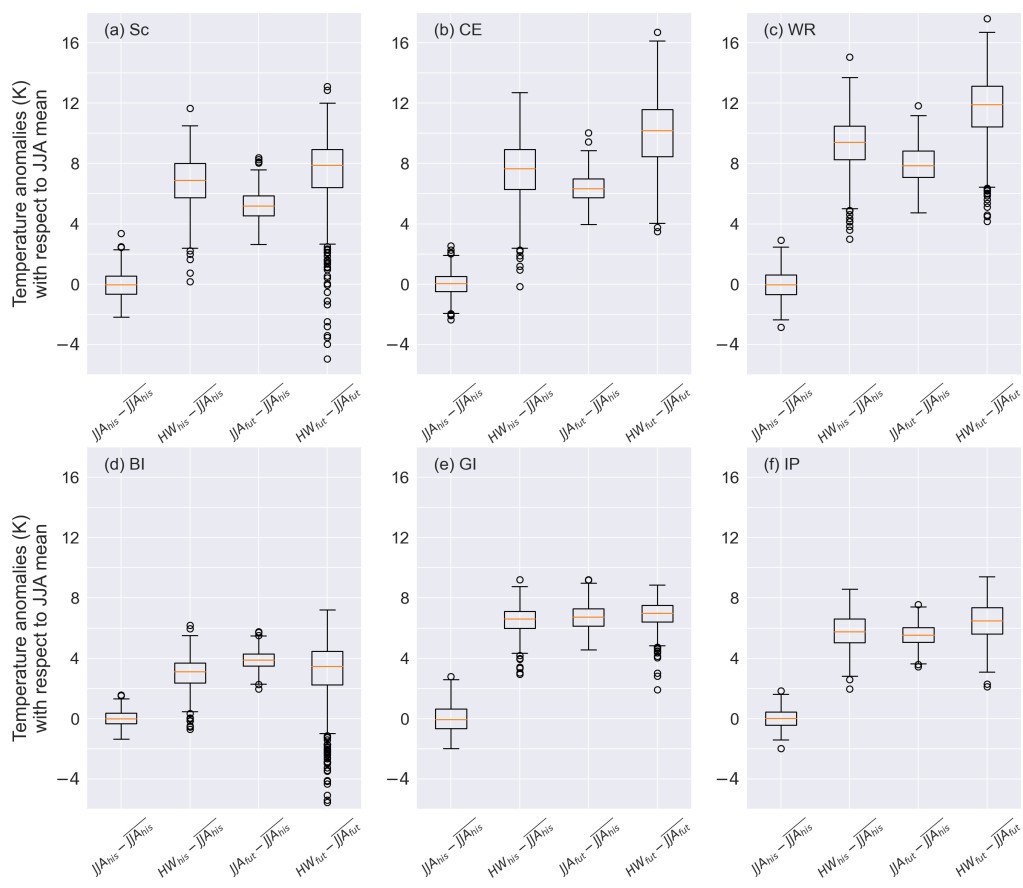

**Figure 2.** Temperature anomalies at 12 UTC (in Kelvin) with respect to the JJA mean temperature in the regions (a) Sc, (b) CE, (c) WR, (d) BI, (e) GI, (f) IP. JJA mean temperatures are calculated over each year of the time slice (historic: 1991-2000; future: 2091-2100) and each ensemble member. The heatwave temperatures are determined as spatial average temperature per region for each heatwave day. Note that no differentiation between grid points over land/ocean are made here. For the first three boxes from the left, temperature anomalies are calculated by subtracting the mean JJA temperature $\overline{\mathrm{JJA_{his}}}$ of the historic time slice. The fourth box shows the future heatwave (HW) temperature $\mathrm{HW_{fut}}$ anomalies with respect to the mean JJA temperature $\overline{\mathrm{JJA_{fut}}}$ of the future time slice. The box spans the 25th and 75th percentile of the data, the horizontal (orange) line inside the box gives the median, and the whiskers are given by 1.5 times the interquartile range; flier points are those outside the whiskers.

the end of July up to mid August (see orange excess in Fig. 4). Moreover, the future frequency distribution narrows with less
heatwave days before mid July. A fit with skew normal distributions shows an increase in skewness and a decrease in scale





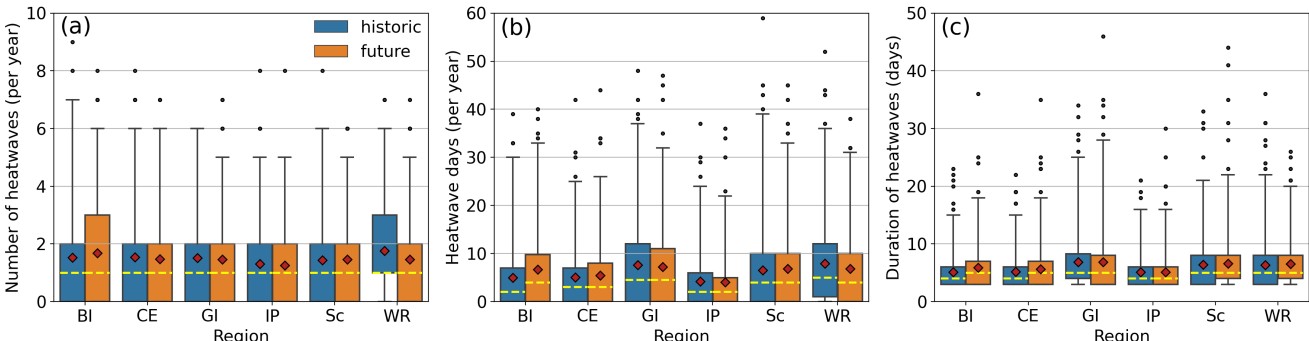

**Figure 3.** Properties of heatwaves in future (orange) and historic (blue) time slices in the six European regions: (a) Number of heatwaves per year and region; (b) heatwave days per year; (c) heatwave duration (in days) with a minimum heatwave lifetime of 3 days.

in the future distribution, i.e., a higher deviation from a normal distribution. The differences in mean and standard deviation between the two periods are statistically significant (see Supplementary Table T1 and Supplementary Figure S1).

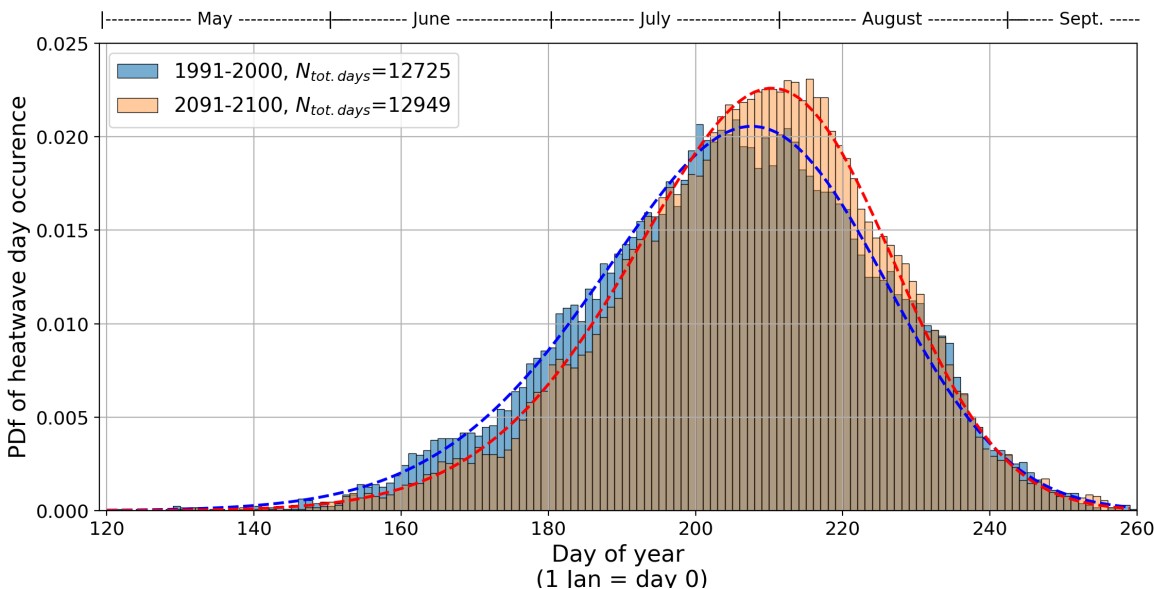

**Figure 4.** Probability density function (PDF) of heatwave day occurrence with respect to the day of the year: blue bars (1991-2000); orange bars (2091-2100); brownish colours show the overlap of the blue and orange histograms. Dashed lines show fitted skew normal distributions; parameters are given in Supplementary Table T1.

A comparison of historic and future distributions of daily temperature maxima on heatwave days reveals that the distributions are primarily shifted towards higher temperatures, but do nearly not change in shape (Fig. 5). However, there is a broadening



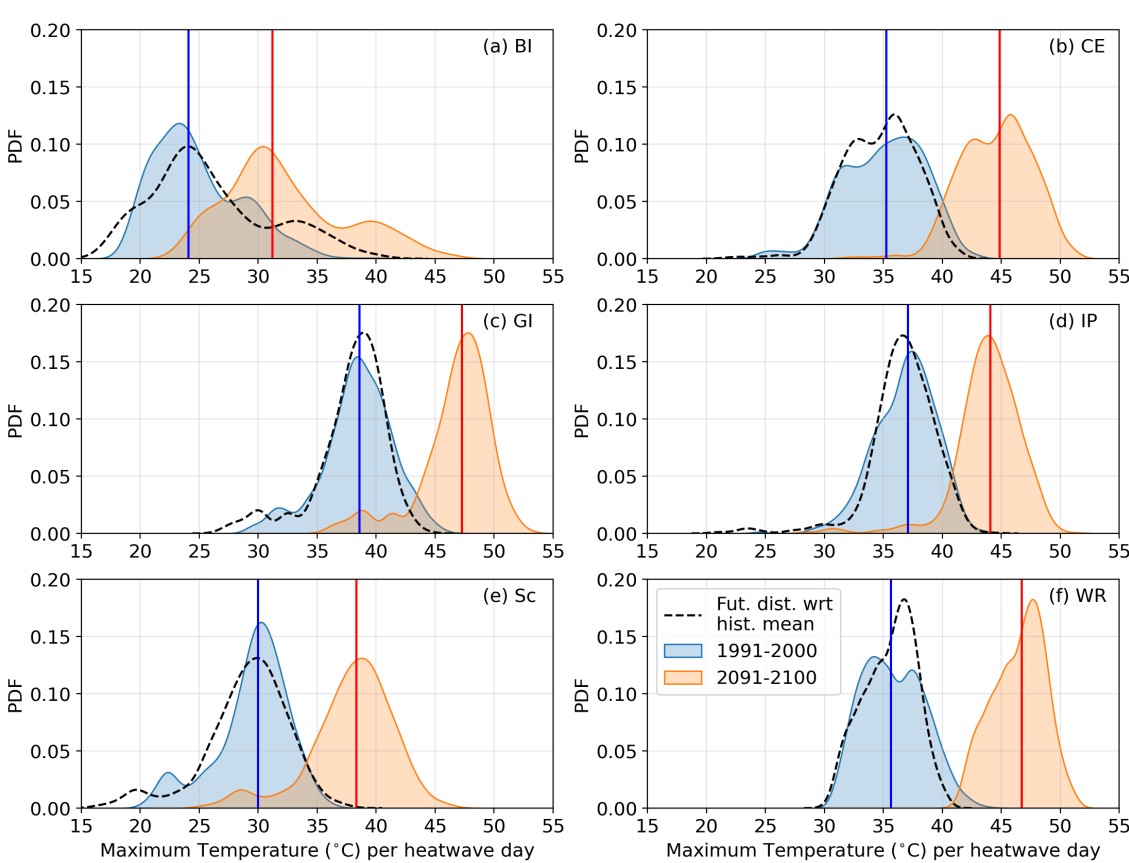

**Figure 5.** Probability density function (PDF, kernel density estimate) of the daily maximum temperature on heatwave days. Note that only the grid point with the maximum temperature is considered per day and region: blue (1991-2000); orange (2091-2100); brownish colours: overlap of both. Black dashed line show shifted future distributions such that the mean corresponds to the mean of the historic distribution.





of the distribution towards higher temperatures in BI (Fig. 5(a)) and a narrowing of the distribution in WR with an increase of
peak probabilities by about 5 percentage points (Fig. 5(f)). Similar results are obtained for the distributions of the HWMId (see
Supplementary Material, see Supplementary Fig. S2).

### 3.2    Composites of 500 hPa geopotential height at first day of heatwaves

A first impression of the dynamical conditions leading to heatwaves and their projected future changes is provided through
composites of the geopotential height fields at the 500 hPa level (Figs. 6 and 7). The composites based on the CESM-LE
simulations for the historic time slice (left sides of Figs. 6, 7) are similar to the results of Zschenderlein et al. (2019), who
studied heatwaves in the ERA-Interim reanalysis data set in the period 1979 to 2016, both with regard to the mean and standard
deviation patterns. In both data sets, the composites are characterized by ridges (positive anomalies) nearby the region affected
by a heatwave with relative low standard deviation compared to the regions up- and downstream of the ridges. However in the
CESM-LE data, the ridge axis is located towards the east of the study areas for some regions (CE, BI and IP), whereas the axis
appears to be more centered in the results of Zschenderlein et al. (2019). In our analysis as well as theirs, the most pronounced
anomaly pattern resembles an Omega block over Scandinavia (Fig 6a). Except for WR and GI, all regions are located west of
the ridge and thus characterized by south-westerly flow in the CESM-LE data. Heatwaves in WR are also associated with an
Omega-like pattern whose center is within the WR region (Figs. 7g). In IP and GI, the heatwaves are associated with extended
subtropical ridges (Fig. 7i,k).

Due to the temperature increase towards the end of the 21st century, the geopotential height of the 500 hPa level is expected
to rise, too[1]. In order to facilitate the comparison between historic and future composites and to identify possible changes
in the dynamic conditions, the mean geopotential height fields are plotted as anomalies on the right sides of Figs. 6 and 7.
In general, the composite patterns remain remarkably similar between future and historic time slices (see also black and red
contours on the right sides of Figs. 6 and 7). The anomaly patterns of all regions point towards a northward shift of the Atlantic
westerly jet stream during future heatwaves (see Figs. 6 and 7). Furthermore, in all regions, the Icelandic low deepens in the
future during heatwave onsets while the Azores high remains more or less unchanged, except for GI (see 7(f) showing a more
pronounced Azores high and approximately unchanged conditions for the Icelandic low). However, the overall changes in the
geopotential height anomalies of the 500 hPa level are relatively small, in the range of about $\pm 40$ gpm. This is about $\pm 0.7\%$ of
the typical height of the 500 hPa level[2]. On the other hand, composites of so many cases tend to smooth differences. In order
to obtain more detailed insights into the relative roles of dynamical and thermodynamic changes for future heatwaves, we thus
investigate these processes using a Lagrangian perspective based on backward trajectories.

---

[1]Under hydrostatic balance, an increase of the mean temperatue of the 1000-500 hPa layer of 5 Kelvin leads to an increase of its thickness by about 100 m.
[2]For comparison reasons: the typical difference in geopotential height between a high and a low pressure system such as the Azores high and the Icelandic
low is on the order of about 10%.



**Figure 6.** Composites of the geopotential height fields at 500 hPa for the first day of heatwaves in different European regions: (a,b) Scandinavia (Sc); (c,d) British Isles (BI); (e,f) Central Europe (CE). The left column refers to the historic time slice (1991-2000) with mean geopotential height (black contours, in gpm) and standard deviation (shaded, in gpm). The right column shows the difference between the historic and future (2091-2100) mean geopotential anomaly fields at 500 hPa. For the calculation of these anomalies, the mean geopotential height at the first day of all heatwaves inside the respective area is first subtracted from the geopotential height composites. The resulting fields are shown by black (historic) and red (future) contours in the right column. The color shading then shows the difference between the two fields (future minus historic); violet (green) shading indicates areas of lower (higher) geopotential height in the future. Relative minima (maxima) of the difference fields (rough estimates) on the right column are labeled by minus (plus) signs.



**Figure 7.** As in Fig. 6, but for (g,h) Western Russia (WR); (i,j) Iberian Peninsula (IB); and (k,l) Greece and Italy (GI).

## 3.3 Origins of air parcel trajectories associated with heatwaves

In order to identify the origins of the low-level air that is associated with heatwaves, we analyze the spatial distribution of
backward trajectories initialized at heatwave locations three days prior to the heat event. These distributions are compared with





typical origins of low-level air masses in summer (JJA). Furthermore, we investigate projected changes of the parcel origins in the future.

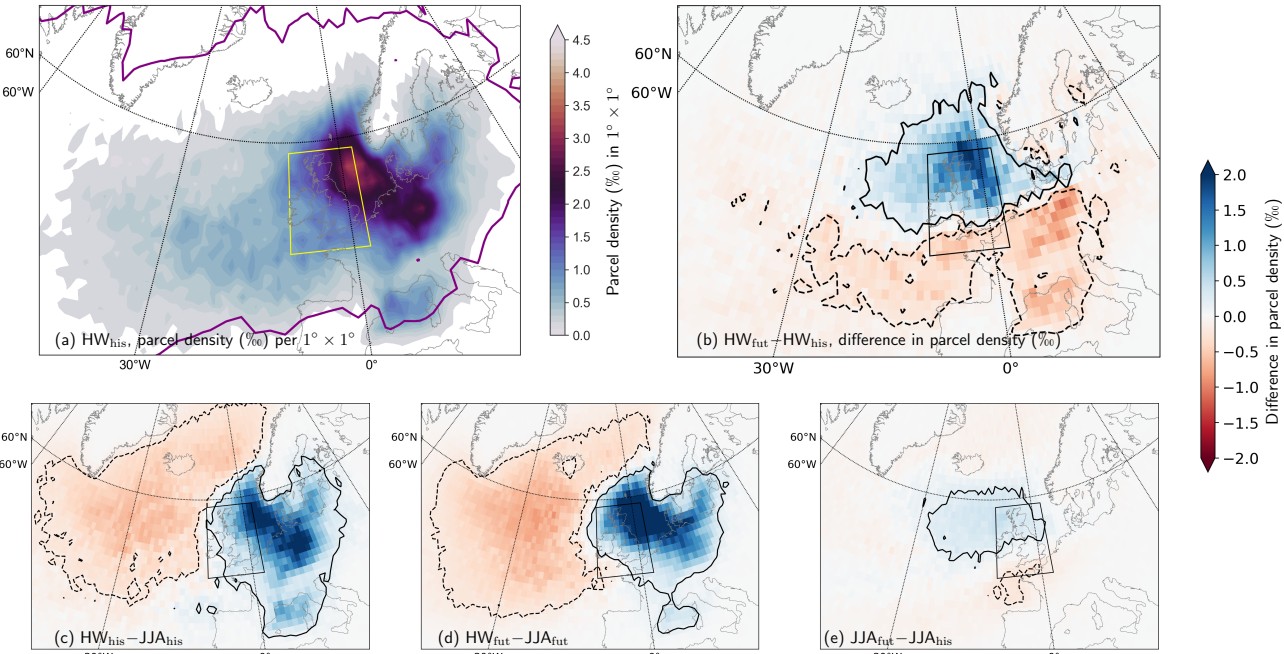

**Figure 8.** Spatial distribution of the trajectories initialized during heatwaves in BI: (a) Density of the parcel trajectories (color-shading) 3 days prior of the heatwaves. The purple contour indicates a parcel density (all trajectories) of 0.1‰ per $1° \times 1°$ seven days prior of the heatwave. (b)-(e) Difference in densities (in ‰) given by red-blue color shading: (b) Difference between future trajectories densities 3 days prior to heatwaves $HW_{fut}$ minus historic ones $HW_{his}$. (c) Difference between historic heatwave $HW_{his}$ and climatological historic JJA trajectories $JJA_{his}$; (d) Difference between future heatwave $HW_{fut}$ and climatological future JJA trajectories $JJA_{fut}$; (e) Difference between future $JJA_{fut}$ and historic, climatological JJA trajectories $JJA_{his}$. Yellow/black polygon shows the study region that was used to initiate the backward trajectories. Solid (dashed) black contour in (b)-(e) marks +0.2‰ (−0.2‰) difference in parcel density.

Figure 8 shows the origins of backward trajectories initialized in BI as well as their differences to the JJA climatology and projected future changes. The trajectory densities for the climatology and future time slices are exemplarily shown in
Supplementary Fig. S3. The majority of the parcel origins 3 days prior to the heatwaves are located east of the study region, mainly over the North Sea and Central Europe (Fig. 8(a)). This is in accordance with the ridge observed in the composite plot (Fig. 6(c)) that suggests reduced westerly and/or occasionally easterly winds towards the BI. A similar result was obtained by Zschenderlein et al. (2019) based on reanalysis data (their Fig. 7), however in their case, the trajectory density is also higher directly in the BI region. The mainly easterly parcel origin is in sharp contrast to the JJA climatological origins of backward
trajectories initialized over BI, most of which are located west of the BI and farther away over the North Atlantic (Fig. 8(c)





and Supplementary Fig. S3. In the future, a shift is projected of the parcel density peak towards the north of the BI and the surrounding ocean and fewer parcels are located southwest and southeast of the BI region (Fig. 8(b)). This shift – but much less pronounced – can also be found in the general JJA climatology (Fig. 8(e)). Together, these changes are associated with an even more pronounced difference between the origins of heatwave and climatological JJA trajectories in the future compared
to the historic time slice (Fig. 8(d)).

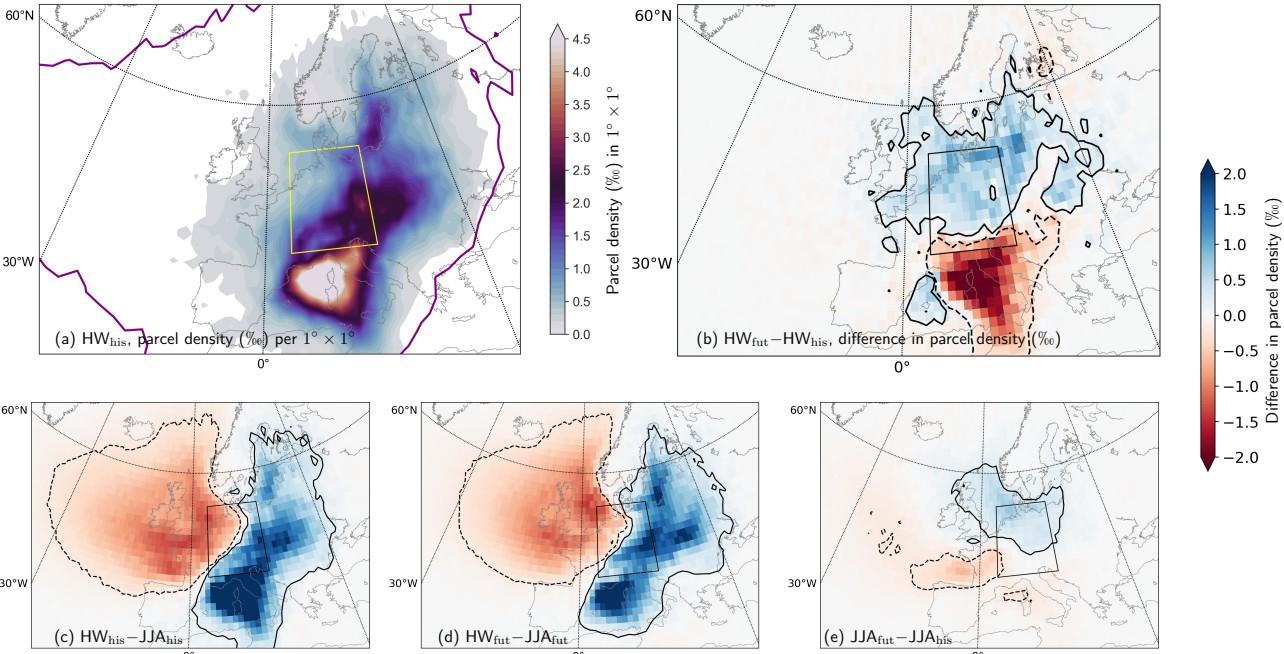

**Figure 9.** As in Fig. 8, but for CE.

In CE (Fig. 9), most of the heatwave related air parcels originate from the Mediterranean Sea south of the CE region 3 days prior to the heat event (Fig. 9(a)). High parcel densities are also found southeast of the CE region as well as generally east of CE. Compared to the results published in Zschenderlein et al. (2019) (their Fig. 7), the peak of the parcel density is shifted more southward in the CESM-LE data. Again, the mainly southerly and easterly parcel origins are in contrast to the general
JJA climatology (Fig. 9(c)), which is associated with typical origins northwest of and partially in the northwestern part of the CE region. For the future, a northward shift of the parcel densities is projected, with a higher density inside the CE region, while the parcel density south of CE becomes smaller (Fig. 9(b)). Projected changes in the JJA climatological parcel origins are generally small, with only a slight increase in parcel densities northwest of the CE region (Figs. 9(e)), suggesting only minor future changes in the general summer circulation over CE.

GI is the only region, where the air parcels associated with heatwaves originate mainly from the western part of the region, with the maximum parcel density over the Mediterranean Sea (Fig. 10(a)). However, there is another less pronounced peak

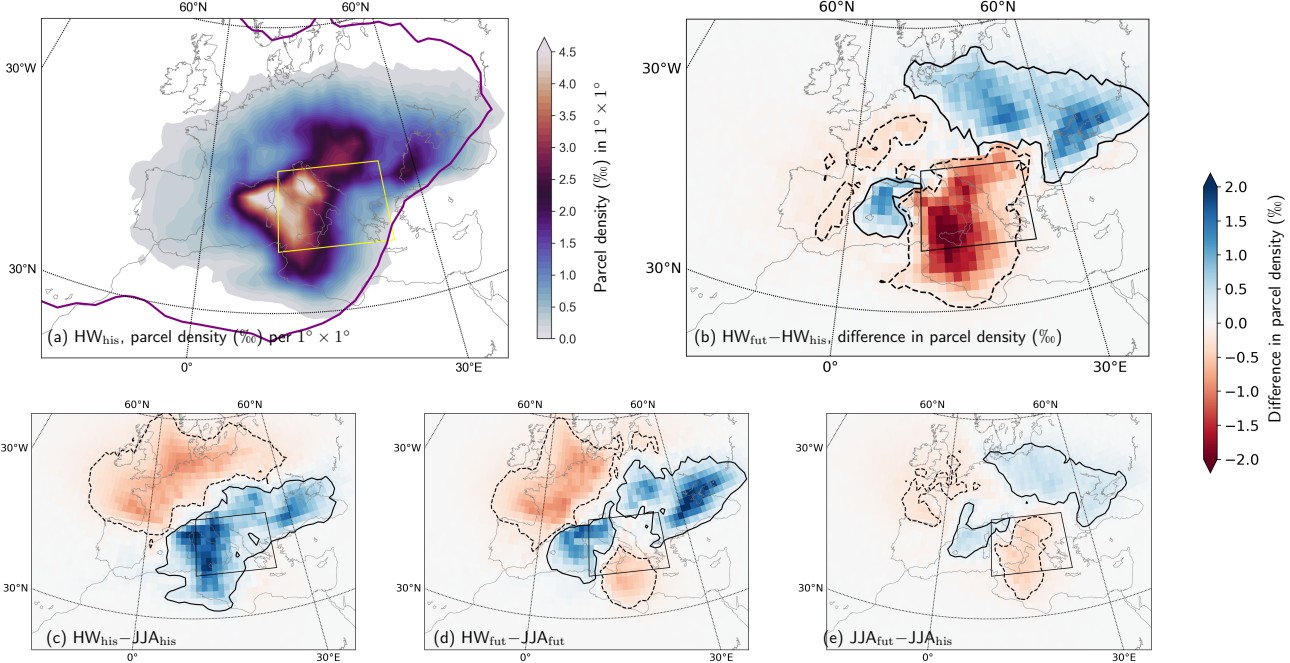

**Figure 10.** As in Fig. 8, but for GI.

in parcel densities north and northeast of the GI region. Overall, these results are similar to those published in Zschenderlein et al. (2019) for the GI region. In the JJA climatology, the peak in parcel origins in the western part of GI is less pronounced, and instead more air parcels have traveled over a longer distance from northern and western Europe towards GI (Fig. 10(c)).

For future heatwaves, the peak in the western part of GI is projected to reduce, and more parcels originate from further north and northeast and from the western Mediterranean Sea east of the Spanish coast (Fig. 10(b)). Again, the difference between historic and future climatologies is only minor and implies a slight northward shift (Fig. 10(e)).

In the IP region, the air parcels associated with heatwave days originate mainly directly from the east of the region in the western part of the Mediterranean Sea, but also from the northern parts of IP with a density peak over the North Atlantic north

of the Spanish coast 3 days prior to the heat event (Fig. 11(a)). This is also in accordance with Santos et al. (2015) who identify easterly/northeasterly flow with hotspots of parcel densities east and northeast of the Iberian peninsula shortly before summer warm events in IP. Moreover, they found anticyclonic flow and high residence times over or near the IP (see Santos et al., 2015, their Fig. 5). Similar results are shown in Zschenderlein et al. (2019), however, their highest parcel densities were rather concentrated inside the IP region and extended only slightly to the north and east. Comparing the heatwave parcel densities

with the JJA climatology, the parcel densities peak farther to the northwest of the region and considerable lower densities are found east of the IP region in the climatology (Fig. 11(c)). In the future simulations, we observe a shift in parcel densities to the



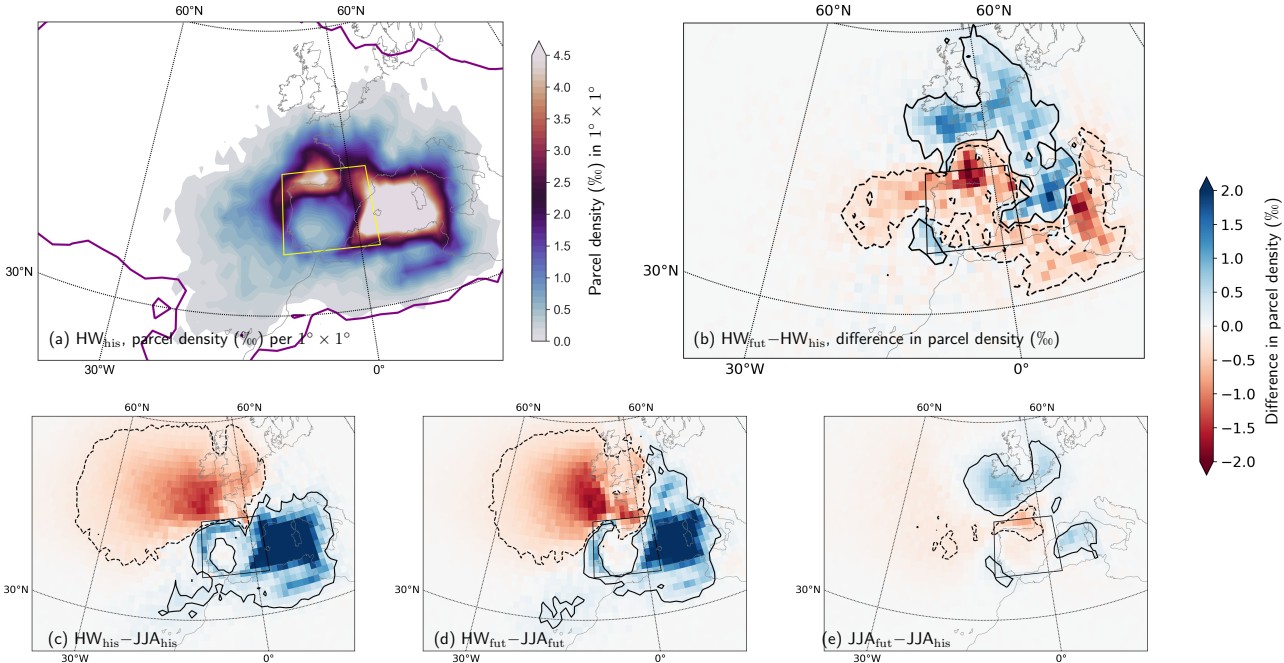

**Figure 11.** As in Fig. 8, but for IP.

north (towards the Channel) and east of the region and an increase in parcel density over the Mediterranean Sea (Fig. 11(b)).
In the future summer climatology, the overall pattern follows that of the heatwaves, but with less distinct changes (Fig. 11(e)).

Air parcel origins of Scandinavian heatwaves are mainly located east and south of the Sc region over the Baltic Sea (Fig.
12(a)). The locations of largest parcel density are comparable to Zschenderlein et al. (2019), but they found higher densities
also inside the Sc region. The general JJA climatology indicates a typical westerly flow in summer in the Sc region, with less
air parcels originating from the south and east compared to heatwave periods (Fig. 12(c)). For future heatwaves, the air parcel
origins are projected to stay relatively unchanged, with only a slight shift towards the north (Fig. 12(b)). Also the summer
climatology in the projected future is almost identical to the present-day distribution (Fig. 12(e)), leading to very similar
differences between heatwave and climatological parcel origins in present-day and future climate (Figs. 12(c,d)).

In WR, the air parcels associated with heatwaves also originate from the east and southeast of the WR region (Fig. 13(a)).
Zschenderlein et al. (2019) found similar results, although their southeasterly peak is even further shifted to the east. Note,
however, that their results for WR were dominated by the exceptionally long-lasting heatwave in 2010. Again, the air parcel
origin during heatwaves is in strong contrast to the JJA climatology, for which the typical flow comes from the northwest and
west. For future heatwave, a slight shift of parcel origins ot the east is projected, and an increase of the parcel density over the
northern part of the Caspian Sea, but overall the changes are relatively small ($< 1‰$) (Fig. 13(b)). Also the changes during of
the climatological air parcel origins are very small, with a slight tendency towards a northward shift (Fig. 13(e)).



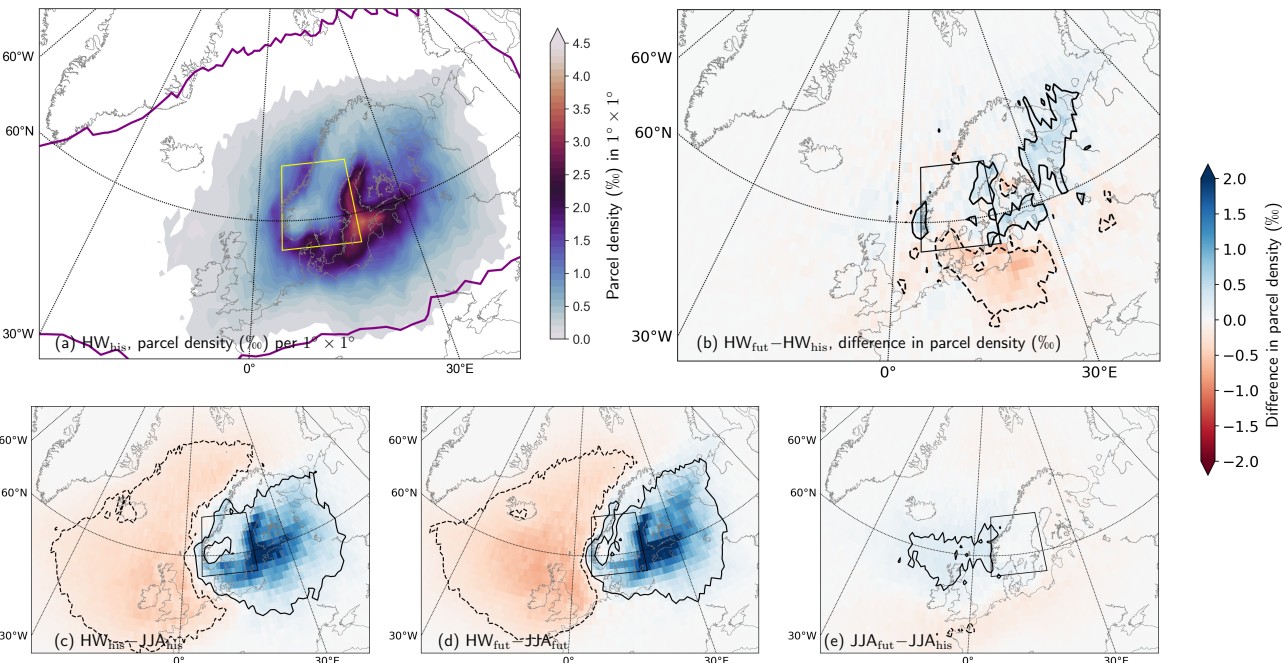

**Figure 12.** As in Fig. 8, but for Sc.

### 3.3.1 Summary: parcel origins

The analysis of air parcel origins in this section has shown that most of the (boundary-layer) parcels associated with heatwaves
stem from inside and/or close to the respective regions 3 days prior to the heat event. This is in accordance with the existing
literature (e.g., Zschenderlein et al., 2019). Compared to the JJA climatology, air parcels associated with heatwaves move from
the east into the respective regions, while the typical summer air streams come from the west. An exception is GI, where the
air associates with heatwaves also comes from the west of the region (western Mediterranean sea). Furthermore, for the CE
region, a considerable amount of air parcels originate south of CE with a maximum over the western Mediterranean sea. In
accordance with the anticyclonic circulation anomalies (see again section 3.2), median parcel trajectories typically follow an
anticyclonic curvature in the last 3-5 days prior to the heat event in the vicinity of the respective regions (not shown). In future
heatwaves, parcel origins generally tend to shift northward in BI, CE, GI, IP. For Sc and WR, projected future changes are
small, with a minor tendency towards an eastward shift. Interestingly, air parcel origins for the entire summer climatology are
not projected to change much, which implies only minor changes in the typical summer circulation.

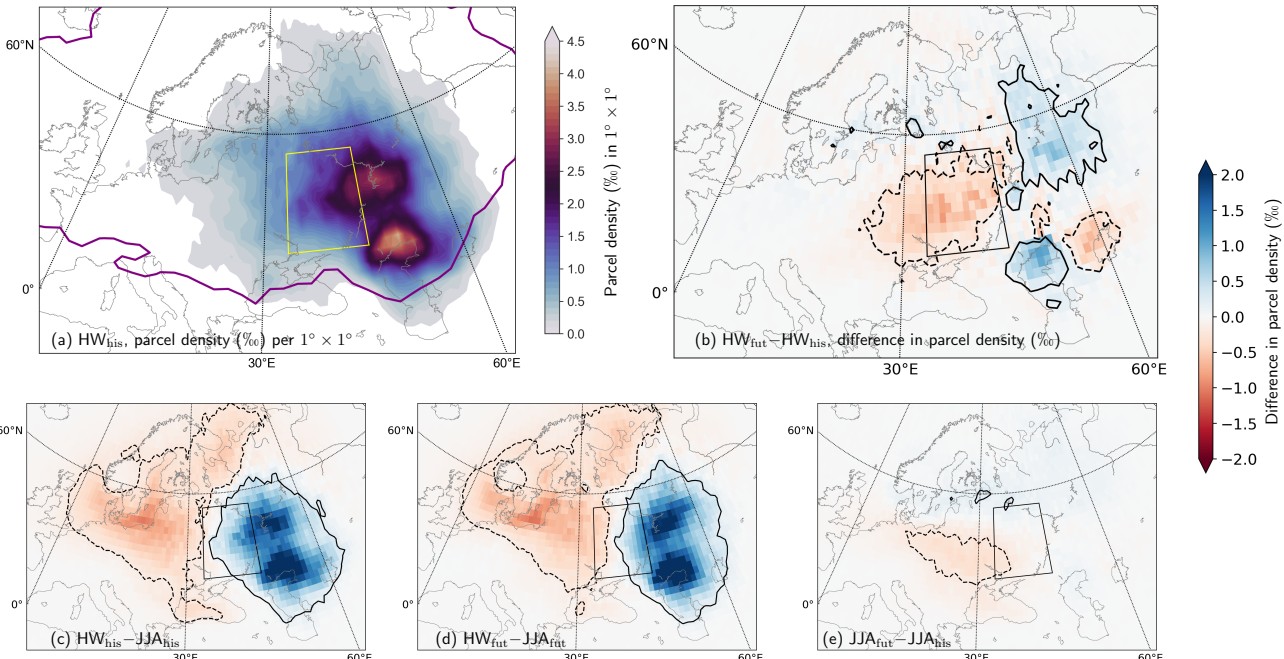

**Figure 13.** As in Fig. 8, but for WR.

### 3.4 Physical characteristics of heatwave trajectories

To study the physical characteristics of air streams associated with European heatwaves, the trajectories are categorized into different clusters as described in section 2.4 (see also Table 2). The majority of all trajectories – independent of the region, time slice and heatwave classification – fall into the cluster categories A or B (Bwd, BSd), which means that along almost all ($> 90\%$) trajectories the temperature increases (Fig. 14). However, the exact partitioning between clusters A, Bwd and Bsd differs between the European regions. We find a higher fraction of air parcels in cluster A, characterized by diabatic cooling, for regions that contain also partially the North Atlantic ocean (BI) or Mediterranean Sea (GI, IP) (see red lines in Fig. 14). While cluster A is least important in WR with only about 10%, the Bwd and Bsd clusters have a larger share in this region. The B clusters are characterized by diabatic heating along the trajectory. Furthermore, air parcels in cluster Bsd strongly descent by more than 50 hPa in the 3 days prior to the starting date of the backward trajectory. The largest fraction of parcels in the Bsd cluster compared to all other regions is found in WR. Compared to the JJA climatology, heatwaves are characterized by a higher percentage of air parcels in the cluster A and a much lower fraction in cluster Bwd in BI, GI and IP, the regions that are affected by the North Atlantic or Mediterranean Sea. In the other regions (CE, Sc, and WR) a much higher percentage of parcels fall into the Bsd category during heatwaves compared to the JJA climatology. Comparing historic and future heatwaves, we observe a decrease in cluster A trajectories in CE, WR and GI, while the fraction of cluster Bsd trajectories increases in the



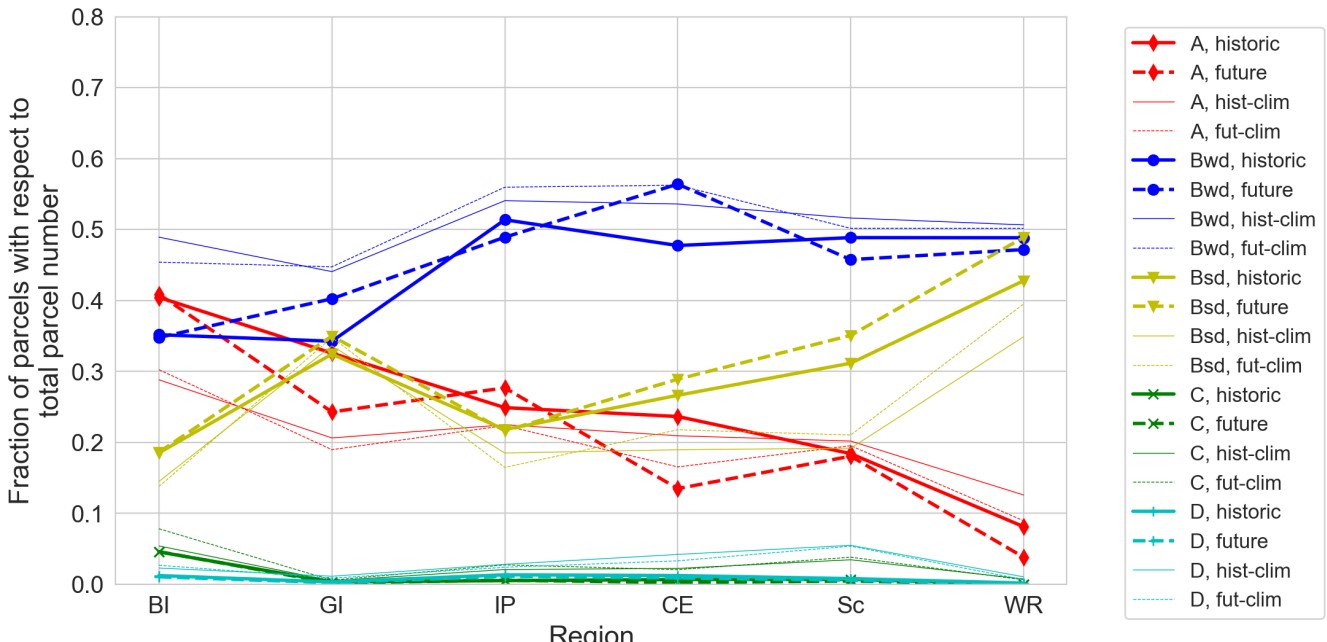

**Figure 14.** Fraction of trajectories that belong to the different clusters A, Bwd, Bsd, C and D introduced in section 2.4 per region and time slice (solid lines belong to historic data and dashed lines to future data). Displayed are the fractions for heatwave trajectories (bold lines) as well as for the general summer (JJA) trajectories (thin lines). The regions are ordered according to the fraction in cluster A. Note that the lines connecting the different regions are only plotted to simplify the identification of the clusters and possess no physical meaning.

same regions and additionally in Sc (yellow lines in Fig. 14). Furthermore, the fraction of air parcels in cluster Bwd increases by about 10 ($\approx 5$) percentage points in CE (GI) (blue lines in Fig. 14).

Fig. 15 shows the median pressure along air parcel trajectories that fall in clusters A, Bsd and Bwd. Cluster A, that is characterized by diabatic cooling, shows the strongest descent over the 10 day period in all regions, although the pressure at $t = -240$h differs between about 700 hPa in BI (Fig. 15(a), yellow lines) and about 500 hPa in WR (Fig. 15(d), yellow lines).

The lowest altitude (highest pressure) 10 days prior to the heatwave is found for the median of cluster Bwd where the parcels first slightly descend for about 7 days until they are close to the surface and then either ascend (in Sc, CE, IP) or remain almost at the same pressure level (in BI, WR, GI, see Fig. 15, blue lines). The median pressure evolution of the Bsd cluster (Fig. 15, red lines) is characterized by an almost constant vertical level for about 6 to 7 days prior of the heatwave at an intermediate pressure of about 750 hPa in most regions. Exceptions are BI with $\approx 820$ hPa (Fig. 15(a)) and WR with about 700 hPa (Fig.

15(d)). The last 3 to 4 days prior to the heatwave event are then characterized by a strong descent in the Bsd cluster with a median change in pressure larger than 100 hPa in all regions. Compared to the typical JJA climatology 10 days before parcels reach the regions, the largest differences in the median pressure are found in regions Sc, CE, WR and IP. In all regions and clusters, except of cluster A in BI and GI, the JJA parcels start from a lower height and their descent is not as strong as in



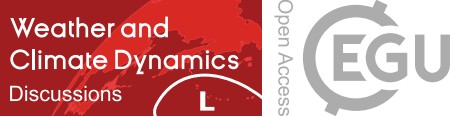

**Figure 15.** Median evolution of pressure (hPa) along the air parcel trajectories in clusters A, Bsd and Bwd in the six European regions. Bold solid (bold dashed) lines represent the air parcels associated with a heatwave event in the historic (future) time slice; accordingly the thin lines show the climatological JJA values.



the heatwave cases. Differences in the median pressure evolution between future and historic heatwaves in the six Euopean

regions are mostly small (bold dashed and solid lines in Fig. 15). The largest changes are projected in WR (cluster A) and IP (all clusters). In WR, we observe a stronger descent in cluster A shortly before the heatwave event and a higher starting point 10 days prior to the heatwave. However, the fraction of parcels in cluster A in region WR is very low ($< 0.05$, see Fig. 14) in the future. In IP (Fig. 15(e)) the median pressure of air parcels in all three clusters is lower for future heatwaves 10 days prior to the event. Moreover, the historic and future JJA pressure medians are almost identical.

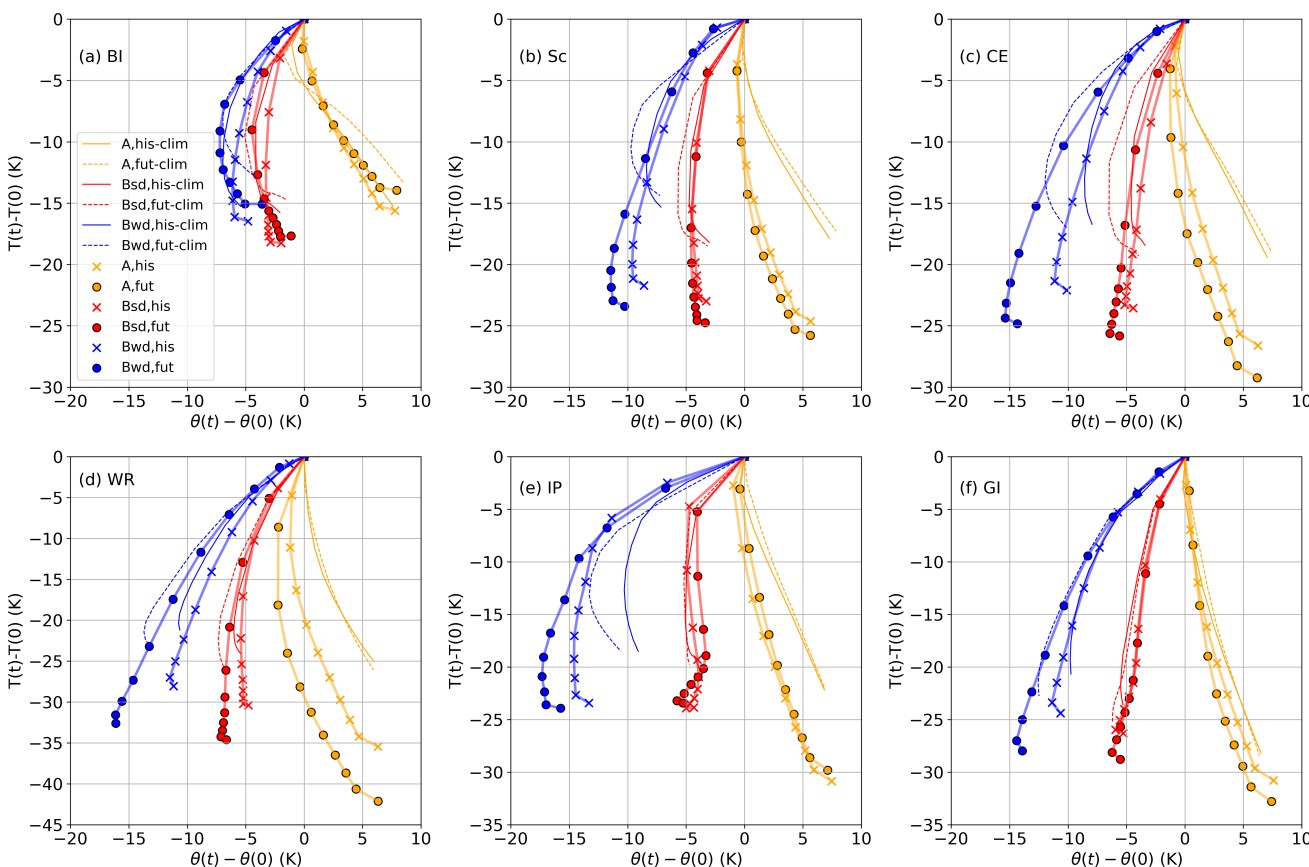

**Figure 16.** Median temperature and potential temperature evolution along the air parcel trajectories for each cluster (A: yellow, Bsd: red and Bwd: blue) relative to the values at the heatwave day in the six European regions, i.e. the point with coordinates (0,0). Dots and crosses indicate the values in intervals of 24 hours, always at 12 UTC. Note the different ranges of the temperature axis (y-axis) for the different regions. Thin lines correspond to historic (solid) and future (dashed) climatological medians.

The median development of temperature and potential temperature along the trajectories is displayed in Fig. 16 for heatwave days as well as for the JJA climatology. Jointly analyzing these two variables allows us to distinguish between future changes in adiabatic and diabatic warming processes (see again section 2.4). Overall, for the JJA climatology (thin solid and dashed



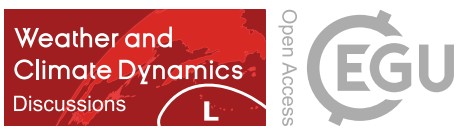

blue lines in Fig. 16), the median temperature and potential temperature evolution stays similar between the historic and future

time slices for clusters A and Bsd, while a stronger diabatic heating (of less than 5 K) is projected for cluster Bwd in most

regions. On the other hand, the evolution differs strongly between heatwave and JJA trajectories, with larger temperature

changes (difference ≈ 10 K) for the heatwave trajectories. In the following, the median distributions associated with heatwave

days in the three clusters as well as their future changes are discussed in detail. In general, cluster A (yellow lines in Fig. 16)

shows the largest temperature increase in the 10 day period prior to the heatwave event (except for the BI). The magnitude of

this increase depends strongly on the region and is largest in WR with about 35 K (historic) to 40 K (future) (vs. 25 K increase

in the JJA climatology) and smallest in BI with about 15 K. Moreover, the potential temperature evolution of cluster A shows

that the air parcels undergo diabatic cooling of about 7 K (up to 10 K in WR) and in some regions diabatic heating right before

the heatwave event (last 24 hours before the event in WR, CE and Sc, Fig. 16(b-d)). Cluster Bsd (red lines in Fig. 16) is

characterized by appoximately adiabatic ($\theta \approx \mathrm{const.}$) warming between day 10 and about day c2 prior to the heatwave in most

regions. In the two clusters, Bsd and A, of heatwave associated air parcels there are only minor differences between future

and historic time slices in BI, Sc and IP, but stronger temperature increases in WR, CE and GI. This temperature increase is

larger (between about 2-5 K in most regions) than the one – if any – observed in the summer climatology. Air parcels in cluster

Bwd (blue lines in Fig. 16) are generally heated (increase of both temperature and potential temperature) over the whole 10

day period with stronger diabatic heating directly before the heatwave event. These Bwd parcels are characterized by weak

descent and hence, they are affected by boundary-layer processes such as surface-atmosphere interactions over a long period

of time. Diabatic heating happens preferentially through surface fluxes, this is why the air parcels with the weakest descent

(that are closer to the surface for a longer period) experience the strongest diabatic heating. An exception is BI, where the Bwd

parcels are first cooled for a few days and then heated in the last days prior the heatwave (Fig. 16). The diabatic heating of the

Bwd parcels over the 10 days is between about 7 K in BI up to about 10 K in WR, CE, IP and GI for historic heatwaves and

becomes even larger in these regions in the future (about 15 K). This increase of diabatic heating for the heatwave trajectories

is similar to the increase projected for the JJA climatology. All together, more pronounced future changes in adiabatic and

diabatic heating (the latter primarily in cluster Bwd) are projected in the regions with continental climate, in particular CE and

WR, compared to those closer to the ocean.

In Fig. 17, an example of the entire distribution of the trajectory properties $\Delta T_{max}$ and $\Delta \theta_{max}$ is plotted for the CE region.

The scatter plot (Fig 17(a)) shows that the majority of the strongly descending air parcels (blue colors) falls in cluster A.

Strongly descending parcels can also be found for large temperature increases and weak diabatic heating in cluster B. Strongly

ascending parcels ($\Delta P_{3d} < -100$ hPa) are connected with strong diabatic heating and weak temperature changes in either

cluster B or C. The majority of all parcels associated with heatwave days in CE are in cluster B with temperature increases

$\Delta T_{max}$ between approximately 20 to 40 K and diabatic heating $\Delta \theta_{max}$ up to about 20 K in the historic time slice (Fig. 17(b)),

and up to about 30 K in the future time slice (Fig. 17(c)). With respect to the thermodynamic properties of the parcels, we thus

observe a future shift towards increased diabatic heating along the parcel trajectory in CE (cluster B, Fig. 17(d)). Additionally,

we find an increase in the number of parcels with a larger temperature increase in the future (also cluster B) and an overall

smaller probability for parcels in cluster A. The distributions of the other European regions, except of BI, are similar to the CE

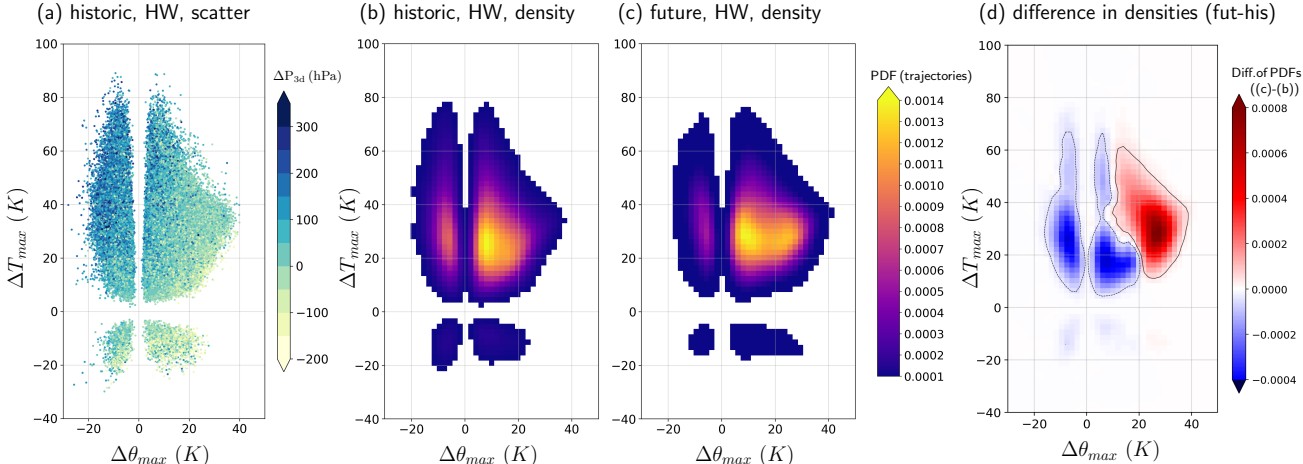

**Figure 17.** Maximum temperature difference $\Delta T_{max}$ along trajectories associated with heatwaves plotted against their maximum potential temperature difference $\Delta \theta_{max}$ for the CE region. (a) Scatterplot with each dot representing the properties of one trajectory; descent/ascent in the three days prior to the heatwave $\Delta P_{3d}$ is indicated by colors; (b,c) Probability density function (Gaussian kernel-density estimate) of trajectory counts (b) in the historic time slice and (c) in the future time slice. Violet to yellow colors show the probability in grid boxes of $2K \times 2K$. Probabilities $< 0.00001$ are omitted. (d) Difference between PDFs shown in (b) minus (c); absolute values of probability differences smaller than $< 0.000001$ are omitted. Solid (dashed) black contour marks a value of $+0.00005$ ($-0.00005$) of the difference in PDFs.

plots, however the details differ from region to region (see 1st column in Fig. 18, 19 and the Supplementary Figs. S4 and S5 for similar plots for the other regions). For example, we observe a higher number of strongly descending parcels in GI in cluster A and an increasing probability of cluster A in IP and BI in the future that is not observed in the other regions. This increase in cluster A is also in accordance with Fig. 14 that showed a higher fraction of that cluster in the future in these two regions.

The general pattern of increasing diabatic heating for medium range temperature increases (up to about 40 K) also occurs in the general JJA climatology (4th column in Figs. 18 and 19, especially in GI, CE, IP, WR). However, compared to the general JJA climatology the heatwave trajectories are shifted towards larger temperature increases ($\max(T) > 20K$) (all regions in clusters A and B, see 2nd and 3rd columns in Figs. 18 and 19). Moreover, the parcel distributions of future and historic heatwaves are additionally shifted towards larger diabatic heating ($\max(\Theta) > 20K$) (all regions except for BI) compared to their respective climatologies.

Trajectory properties are displayed as boxplots for the respective clusters in Figs. 20 and 21 in order to further summarize and compare future and historic heatwaves as well as climatological JJA conditions. BI is the only region, where we see no difference in the thermodynamic properties such as maximum increase in temperature (Fig. 20(c)), maximum increase in potential temperature (Fig. 20(b)) and maximum descent in the 3 days prior to the initiation of the trajectory (Fig. 20(a)) between the heatwaves and climatologies. However, we observe large differences in the potential temperature distribution 7







**Figure 18.** Maximum temperature difference $\Delta T_{max}$ along trajectories plotted against the maximum potential temperature difference $\Delta\theta_{max}$ for (a)-(d) BI; (e)-(h) GI and (i)-(l) Sc. Displayed is the difference between PDFs (as in Fig. 17(d)). Different regions are shown in rows (four figures per region) and several different plots are shown in columns: (1st column) Difference between future and historic heatwave distributions; (2nd column) historic heatwaves PDF minus historic climatological JJA PDF; (3rd column) future heatwaves PDF minus future, climatological JJA PDF; (4th column) difference between future and historic climatological JJA PDFs. The probability has been calculated in grid boxes of $2K \times 2K$ and absolute values of probability differences smaller than $< 0.000001$ are omitted. Solid (dashed) black contour in (d) marks a value of $+0.00005$ ($-0.00005$) of the difference in PDFs.



**Figure 19.** As in Fig. 18, but for (a)-(d) CE, (e)-(h) IP and (i)-(l) WR.







**Figure 20.** Trajectory properties of clusters (defined in section 2.4) calculated at heatwave days in comparison to climatological (JJA) values in different European regions: (a)-(d) BI, (e)-(h) Sc, (i)-(l) WR. Top row shows the difference in pressure $\Delta P_{3d}$ three days prior to the heat event/prior to arrival in the target region; second row from top shows the maximum change in potential temperature $\Delta \theta_{max}$ along the trajectory; third row from top is the maximum change in temperature $\Delta T_{max}$ along the trajectory, and bottom row is the potential temperature seven days prior to the arrival in the respective region. *Historic* (violet, third boxes) and *future* (red, fourth) boxes refer to air parcels started at heatwave days, *historic_clim* (green, first boxes) and *future_clim* (yellow, second) boxes show the JJA climatology.





**Figure 21.** Same as in Fig. 20 but for the other European regions: (a)-(d) CE, (e)-(h) GI, (i)-(l) IP.





days (−168 hours) prior to the initiation of the backward trajectory (Fig. 20(d)). There is nearly no overlap of the interquartile ranges between the heatwaves and the respective climatologies as well as between historic and future heatwave air parcels.

This implies that the air originates from higher altitudes with larger potential temperature values and/or from a region with higher than usual temperatures. For BI, in particular the latter is important (enhanced transport form continental regions in the east, which are particularly warm in summer, see again Fig. 8), as the pressure level at −168 hours does not differ strongly between heatwaves and climatology (see again Fig. 15). Similar results with respect to the potential temperature 7 days prior to initialization are found in all other European regions (Figs. 20, 21(d),(h),(l)). Nevertheless, for the other regions also variations

in the pressure at −168 play a more important role (Fig. 15). Moreover, in the other European regions heatwave trajectories differ from their climatological counterparts also with respect to the other thermodynamic properties shown in Fig. 20. For example, air parcels in cluster A (e.g., in Sc and WR, see Fig. 20(e),(i)) experience stronger subsidence than the climatology. This difference in subsidence in cluster A even increases in the future heatwaves (red boxes), in particular for WR (Fig. 20(i)). This larger subsidence coincides with a larger increase in temperature along the trajectories (e.g. Fig. 20(g),(k)) and hence this

increase can be attributed to the adiabatic compression of the parcels that overcompensates the diabatic cooling.

### 3.4.1 Summary: characteristics of air parcels

In summary, future heatwaves are projected to intensify due to (i) increased diabatic heating (all regions, except for BI), (ii) increased descent and adiabatic warming (mainly regions WR, CE, Sc) and (iii) warmer air parcels already prior to transport to the heatwave region (all regions). Point (iii) corresponds to the overall increase of summer temperature, and a similar warming

can thus be observed also for other trajectories not necessarily related to heatwaves (JJA climatology). In contrast, points (i) and (ii) can be specific or amplified for heatwaves, such as the enhanced diabatic heating in WR, CE, GI and SC in cluster Bwd and the enhanced descent and adiabatic warming in cluster A. These processes can thus lead to an intensification of heatwaves beyond the average summer warming.

## 4 Discussion and Conclusions

In this work, we have analysed dynamical and thermodynamical mechanisms associated with European heatwaves and their projected future changes based on Lagrangian air parcel trajectories. In order to properly cover the influence of natural variability, heatwaves are identified in a large ensemble of climate simulations (CESM-LE data with 35 members, Kay et al., 2015) for two time-slices, 1991-2000 (historic) and 2091-2100 (RCP8.5), based on the Heat Wave Magnitude Index daily (HWMId) – a percentile-based method introduced by Russo et al. (2015). This percentile-based index allows for a comparison of heat-

waves in both time slices with each other since it relates the heat event to the underlying climatology and hence accounts for an expected anthropogenic temperature increase in the future. Moreover, the Lagrangian analysis permits to investigate the origin of near-surface air masses associated with heatwaves and to study the thermodynamic processes along their transport pathways. Our main goals have been to investigate the role of dynamic and thermodynamic aspects of heatwaves and how these





might change in the future. In the following, we summarize and briefly discuss our main results, also in comparison to recent

literature.

(1) **Minor changes in the general heatwave characteristics in most European regions:** The percentile-based identification of heatwaves via the HWMId is – as expected – remarkably robust in both time slices with respect to the number of heatwave days and the heatwave duration in all regions, except for BI, where an increase in the number of consecutive heatwave days is projected. This is also in accordance with Vogel et al. (2020) who state that "for fully moving thresh-

olds, no or only very few significant changes in heatwave characteristics with increasing warming levels are projected". Additionally, we observe a significant, albeit small, shift of the heatwave peak towards the end of July and beginning of August in BI, CE and IP and a significant, but small, narrowing of the heatwave season in CE, GI and WR.

(2) **Origins of heatwave surface-near air:** In general, the core origin regions of the near-surface airmasses that contribute to the heatwaves lie to the east of the respective region 3 days prior to the heatwave. More precisely, these core regions

are located southeast of the northerly sub-regions and northeast of the southerly subregions (and in GI even partially to the west). This is in broad agreement with Zschenderlein et al. (2019), who, however, found higher trajectory densities within the region affected by the heatwave itself. Nonetheless, our results also indicate that the air is located close to the analysed region where the heatwave occurs and is not advected from remote (warmer) regions. This is in line with Bieli et al. (2015), who also found short transport distances for heatwave-related trajectories. In comparison to the

climatological transport pathways of near-surface air in summer, the heatwave air parcels origin closer to the regions of interest a few days prior to the heat event and travel slower. Furthermore, the model projections indicate a northward shift of heatwave-associated air parcel origins 3 days prior to the heatwave in the simulated future climate, with higher densities mainly north of and inside (BI, CE) the regions and reduced densities to the south of (and inside for IP, GI) the respective regions. This northward shift in the future can also be seen in the JJA climatology of most regions, however

with smaller magnitude, such that the changes in the climatological parcel origins are mostly very small.

(3) **Projected changes in dynamic and thermodynamic air parcel properties relevant for future heatwaves:** Our main findings show a diverse picture of changes in future heatwave characteristics across Europe. In some European regions such as Sc, CE and WR, we find a stronger temperature increase at heatwave days than would be expected because of the increase of mean summer temperature alone. This indicates that additional dynamic or thermodynamic processes amplify

the intensification of heatwaves in some parts of Europe. Indeed, in these regions the subsidence during heatwaves is projected to increase in the future simulations, especially in the last 2-5 days prior to the arrival in the heatwave region, which is associated with amplified adiabatic warming. Moreover, in all regions except for BI the diabatic heating is projected to intensify. The latter affects air parcels located near the surface, and such boundary-layer diabatic temperature changes are driven by sensible heat fluxes that are enhanced when heatwaves co-occur with larger soil moisture deficits,

especially in plain regions (Stéfanon et al., 2014; Schumacher et al., 2019). During heatwaves, the land surface can be expected to become drier in the future (Seneviratne et al., 2010), thus the amplified diabatic heating may be due to an increase in the Bowen ratio, as also suggested by, e.g., Rasmijn et al. (2018) for WR.





Finally, we want to add some final remarks with regard to the recent summer 2022 heat records. On 19 July 2022, Great Britain reported 40.2°C at Heathrow Airport, London (Carrington, 19-Jul-2022 (online). In the CESM-LE simulations pre-
sented her, the BI distribution shows maximum temperatures of up to 37°C in the historic time slice (1991-2000) and up to about 48°C in the future time slice (2091-2100). Assuming a linear increase, Heathrow's temperature record lies at the upper end of the expected range for the 2020s, which is also in accordance with the work of Christidis et al. (2020), who showed that events like the heatwave in July 2022 are possible but rare and will increase towards the end of this century. This also indicates that we are right on the track of the RCP8.5 scenario. Moreover, this suggests that the CESM large ensemble data seems
to provide a realistic representation of possible maximum temperatures and our results provide a guidance for the upcoming heatwave characteristic that could be used to improve adaptation measures for future heat events.

In summary, we have identified thermodynamic as well as dynamic contributions that are relevant for the future amplification of heatwaves in different European regions. In particular for Scandinavia, Central Europe and Western Russia, these processes lead to an increase of heatwave temperatures beyond the mean summer warming. Part of this amplification is due to increased
diabatic heating in the boundary layer, which is likely linked to surface drying and enhanced sensible heat fluxes. Additionally, the excess heating is related to altered atmospheric dynamics, in particular enhanced descent of the air masses leading to stronger adiabatic warming in the simulated future climate, which is in some contrast to previous findings by Vogel et al. (2020) who suggested that changes in dynamics should be of minor importance.

The presented work provides a comprehensive overview of the range of future changes in heatwave characteristics in different
European regions from a Lagrangian point of view and can serve as a starting point for further research. Nevertheless, the topic still needs further investigations, e.g., with respect to the robustness of the findings in other climate models.

*Code and data availability.* The code of the CESM version 1 that was used for the Large Ensemble simulation is available from https://www.cesm.ucar.edu/models/cesm1.0/ (last access: March 2022; NCAR, 2020). The code of the trajectory model Lagranto is available from https://iacweb.ethz.ch/staff/sprenger/lagranto/download.html (last access: July 2022). Model output is available upon request from the
authors.

*Author contributions.* LS and SP designed the study. LS produced the results and figures. LS and SP discussed the results and both contributed to the writing.

*Competing interests.* S.P. is a member of the editorial board of Weather and Climate Dynamics. The authors have no other competing interests to declare.



*Acknowledgements.* The authors would like to thank the HPC Service of ZEDAT, Freie Universität Berlin (HPC system Curta, see Bennett et al., 2020, for details), for providing computational resources. Furthermore, we are grateful to Urs Beyerle (ETH Zurich) for performing the CESM-LE re-runs.



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
