# Peer review of "European heatwaves in present and future climate simulations: A Lagrangian analysis"

_Weather and Climate Dynamics, 2022_

## Author Comment (AC1)

**Responses to the comments of Reviewer 1, 2 and the Editor**

by Lisa Schielicke and Stephan Pfahl

We would like to thank both reviewers and the editor for their helpful comments. This document is structured as follows: We address all comments of reviewers 1 and 2 in section 1 and 2, respectively. In section 3, we provide answers to the editor's comments. Our responses are printed in blue and labeled by the abbreviation AC (author comments). The referees and editor's comments are given in black and italic and labeled by RC1, RC2 and EC1, respectively.

**1 Comments of Reviewer 1:**

**1.1 Reviewer 1 – General comments:**

RC1: *This manuscript/preprint is devoted to the analysis of European heatwaves under recent past (1991-2000) and future (2091-2100, RCP8.5) climatic conditions. A 3-D Lagrangean description (10-day backward trajectories) of the air mass stream flows underlying the occurrence of percentile-based heatwaves (Heat Wave Magnitude Index daily), with relative and time-dependent thresholds, in gridpoints within six target regions over Europe (following Zschenderlein et al., 2019) is carried out. A 35-member/initialization ensemble, generated by the Community Earth System Model (CESM1), is used for climate change impact assessments.*
*Overall, the topic of research is quite pertinent in the ongoing climate change context, with major and unprecedented heatwaves hitting many regions worldwide. Understanding their driving (dynamical and thermo-dynamical) mechanisms is of utmost relevance for the scientific community and society in general. The data and methods used are adequate for the study's purposes. State-of-the-art datasets are selected. A sufficient literature review is provided to the readers. The results are scientifically sounding and in agreement with previous research. The text is easy to follow, with enough explanation of the methodologies and findings. The figures are of good quality but too much. Therefore, I recommend the acceptance of this manuscript after some minor revisions that are outlined in the specific comments below.*

AC: Thank you very much for your helpful comments! We will reduce the number of figures and move some of the figures to the supplementary material.

**Reviewer 1 – specific comments:**

RC1: *In the abstract: please state that you have used RCP8.5 in your climate change projections.*

AC: We will add a note to the abstract.

RC1: *Section 2.1: Although using a single general circulation model (GCM), i.e. CESM1, is understandable due to the lack of other similar data sources, this is indeed an important limitation of the study that should be stressed. Furthermore, the potential implications of using a single GCM in the results should also be more deeply discussed in Section 4. Only a short sentence is related to this point in the last paragraph of Section 4.*

AC: You are right, we will discuss this limitation in more detail. As we need high-resolution output on model levels for our trajectory calculations, these analyses cannot easily be performed based on other model simulations, e.g., from the CMIP6 archive. Nevertheless, our comparison with the results of Zschenderlein et al. (2019) based on ERA Interim reanalysis data indicate that CESM1 captures the basic dynamics of European heatwaves reasonably well, which gives confidence in the corresponding future projections.

RC1: *Section 2.1: "Re-runs of the simulations have been performed for two 10-year time slices. . . ". Please specify who did these new simulations and if they are available for upcoming studies.*

AC: The re-runs were performed by Urs Beyerle (ETH Zurich), as specified in the Acknowledgment section. The data set is very large and can therefore not be easily made publicly available. We would like to encourage interested readers to contact us in case of questions, and we'd be happy to provide the model output on an individual basis.

RC1: *Section 2.3: The heatwave definition based on the 90th percentile of the daily maximum temperature within a 30-day running window is a reasonable choice. However, something should be mentioned regarding the possible implications of having a higher threshold (e.g., 95th percentile) or a different window length. A sensitivity analysis could be useful to clarify this issue and improve the robustness of the results.*

AC: The heatwave index applied in this study is very well established and has been widely used. The choice of the parameters (90th percentile level, the 5% coverage of land grid points and the ≤3 days) follows Zschenderlein et al. (2019) to allow for a direct comparison with their results based on reanalysis data, which, in our opinion, is an important part of our study. Increasing the percentile threshold and the minimum life time will naturally lead to lower numbers of heatwave cases. Of course it would be interesting to look at the most intense heatwaves and how these will change in the future, which would be an interesting follow-up study. Nevertheless, rerunning the analysis for different thresholds would be a lot of effort (e.g., the present setting requires computing more than 1 million trajectories for the heatwaves), would make an already long paper even longer, and, in our opinion, thus is beyond the scope of this study.

RC1: *Section 2.3: The choice of having different thresholds for each time slice is plausible. Nonetheless, this choice deserves further explanation. In Section 1 this issue is already mentioned, but it would be useful to address this point here as well.*

AC: Thanks! We will address this topic in more detail in section 2.3.

RC1: *Line 114: Please explain "annual maximum temperature". Have you also applied the 30-day running window to the heatwave magnitudes? This is important to understand what you mean by "The latter criterion (Md ¿ 0) also makes sure that heatwaves occur during the warmest time of the year, typically the summer months". The heatwave definition deserves a better explanation for a reader not familiar with this concept.*

AC: The annual maximum at a grid point is just one value per year (i.e. the maximum of the 2m temperature). The percentiles are then calculated from the 350 values in each time slice (35 members times 10 years). Since the annual maximum temperature typically occurs in the warm season, $M_d > 0$ is connected to the warmer season, too. We will clarify this in the text.

RC1: *Equation 2: I would say "isobaric advection" rather than "advection", as vertical advection corresponds to the adiabatic expansion/compression term.*

AR: Thanks for your thoughts. However, we disagree in this point: The advection term in eq. 2 represents the full, three-dimensional advection, not just the isobaric one. We add the derivation here for clarification. Note that in the original manuscript there has been a sign error in this equation, which we will correct in the revision:

$$du = c_v dT = -p d\alpha + \delta q \tag{1}$$
$$= -d(p\alpha) + \alpha dp + \delta q \tag{2}$$
$$= -d(RT) + \alpha dp + \delta q \tag{3}$$
$$(c_v + R)dT = \alpha dp + \delta q \qquad \qquad |R = c_p - c_v \qquad . \tag{4}$$
$$c_p dT = \alpha dp + \delta q \tag{5}$$
$$dT = \frac{\alpha}{c_p} dp + \frac{\delta q}{c_p} \qquad \qquad \left| \frac{d}{dt} \right. \tag{6}$$
$$\frac{dT}{dt} = \frac{\alpha}{c_p} \underbrace{\frac{dp}{dt}}_{=\omega} + \frac{1}{c_p} \frac{\delta q}{dt} \tag{7}$$
$$\frac{\partial T}{\partial t} + \vec{v} \cdot \nabla T = \frac{\alpha}{c_p} \omega + \frac{1}{c_p} \frac{\delta q}{dt} \tag{8}$$

where $\alpha = \rho^{-1} = V/m$ is the specific volume; $T$: temperature, $p$: pressure, $R$: gas constant of dry air, $c_v$: specific heat capacity at constant volume, $c_p$: specific heat capacity at constant pressure, $\delta q$: heat, $u$: internal energy, $t$: time.

RC1: *Figs 8-13: I suggest reversing the colour scale in the panels with the differences in parcel density (b-e), as reddish (bluish) colours are more commonly associated with positive (negative) anomalies.*

AC: Thanks for your comment. We will replot the figures indicating the statistical significance (see below) and following your advice with regard to the color scale.

RC1: *The number of figures is excessive. I suggest using supplementary material for some of them and shortening part of the text, namely sections concerning Figs 2-7. The authors may also choose to present the results only for some of the target regions, leaving the other figures as supplementary material.*

AC: Thanks, you are right. We decided to reduce the number of figures and shift some of them to the supplementary material, e.g. Fig. 4 and 5. We will further recombine parts of Figs. 8-13 to two figures with focus on the heatwaves and their future changes. The additional information on the comparison to the climatology will be moved to the supplementary material. We plan also to reduce the content in Figs. 18/19 and 20/21.

**Reviewer 1 – Technical corrections:**

RC1: *Lime 114: Delta y instead of Delta t in the 25th percentile.*

AC: Thanks, we will change this.

RC1: *Figure 2: no reference is made to the second box of HWhist.*

AC: Thanks, we will rewrite the caption to address all boxes correctly.

RC1: *Line 315: awkward sentence. Please revise.*

AC: We will rewrite the sentence.

RC: *Line 465: "presented her". Please correct.*

AC: We will correct to "presented here".

**2  Reviewer 2 – Comments:**

RC2: *This manuscript addresses an important topic in a novel and informative way. The presentation is clear and the methods appear sound. I have a couple of comments:*
*It is difficult to assess whether the changes projected by the model are significant. Some measure of statistcal significance of the change values is needed for at least some of the analyses. The maps in Figures 8-13 would especially benefit from this*

AC: We will perform t-tests to determine the significance of the differences in means (box plots). Moreover, we will use a bootstrapping method to determine the significance in differences between future and historic parcel origins (Figs. 8-13 in the original manuscript).

RC2: *Figures 15-21 could be condensed to a subset of the figures. Part of the challenge is the multiple subregions, which I do see value in including in each analysis. However, I don't think all of the information presented in these figures is necessary as the results tend to show similar conclusions in diferent ways.*

AC: Thanks, you are right. We decided to reduce the number of figures and shift some of them to the supplementary material, e.g. Fig. 4 and 5. We will further recombine parts of Figs. 8-13 to two figures with focus on the heatwaves and their future changes. The additional information on the comparison to the climatology will be moved to the supplementary material. We plan also to reduce the content in Figs. 18/19 and 20/21.

RC2: *The authors use the 2022 London heatwave to conclude that warming is following the RCP8.5 trajectory. I would suggest mentioning some caveats here. For one, there is growing evidence that emissions are deviating from RCP8.5. Also, using a single event could be a little misleading on assessing a trend. I do think putting this event into context, however, is helpful.*

AC: Thanks, you are right. We will rewrite the part concerning the 2022 London heatwave and remove the specific reference to the RCP8.5 scenario, also because up to about 2030 to 2040 all scenarios show similar ranges of temperature change. The main point of this short paragraph is to give an example showing that heatwaves are already changing in the direction as projected for the end of the century by global climate models, which may help readers to get a more concrete idea of such changes to come.

**3 Editor – Comments:**

EC1: *I would like to thank the two reviewers for their constructive comments, which I hope will be helpful for the authors. I note that both reviewers have suggested that the number of figures could be reduced, so this would be good to attempt in a revised version. I also agree with reviewer 2 that some measure of significance would be beneficial to improve our confidence in the projected changes.*

AC: Thank you very much! The comments of the reviewers are indeed very helpful. We will reduce the number of figures, e.g. we will move Figs 4 and 5 to the supplementary material; the information on Figs 8-13 will be reduced to focus on the heatwaves and the additional information on the climatology will be moved to the supplementary material. This will reduce the number of figures from 6 to 2. We also try to reduce the content of Figs. 18/19 and 20/21 by selecting only some of the regions and moving the rest to the supplementary material. With respect to the significance, we will use a bootstrapping method to determine the significance in differences between future and historic parcel origins (Figs. 8-13 in the original manuscript). Additionally, we will use t-tests to determine the significance of the differences in means (box plots).

EC1: *I also have one additional comment. Although this is a widely-used model it would still be good to touch on the issue of model evaluation, since all of the results rely on the fidelity of this model. How similar are the detected events in this model to those in observations and/or reanalyses, and how does the model perform in evaluation / validation exercises in aspects relevant to European heatwaves? Hopefully this could be addressed using existing literature rather than new data analysis.*

AC: Our methodology closely follows the study of Zschenderlein et al. (2019), who performed a similar analysis based on ERA-interim reanalysis data. This has enabled us to compare our results on heatwaves in the historical time slices with theirs, which has already been described at several places in the original manuscript. As discussed there, there is a reasonable agreement with regard to general heatwave statistics (see section 3.1 of the original manuscript), composite patterns (section 3.2), and air parcel origins (section 3.3). For the latter, Zschenderlein et al. (2019) found the maximum of parcel density three days prior to the heat event onset mainly inside of the regions of interest, while in CESM it typically is slightly displaced, but still in close proximity. Nevertheless, the overall agreement makes us confident that we can also trust the future projections of European heatwave dynamics of CESM. This will be emphasized more explicitly also in the conclusions section of the revised manuscript. Furthermore, Schaller et al. (2018) show that the relationship between European heatwaves and blocking (which is central for the circulation associated with heatwaves) is properly represented in CESM-LENS, also in comparison to another climate model ensemble. Nevertheless, CESM is not able to reproduce events as extreme as the Russian heatwave in 2010, which may point to model biases in the representation of very extreme events. At the same time, Fischer et al. (2021) find, in a gridpoint based analysis of warm extremes, that the probability for "record-shattering" events, i.e., events larger or equal to twice the standard deviation, for the large ensembles of the CESM-model family is at the higher end of the model range compared to CMIP5 and CMIP6 ensembles (note: their ensemble NCAR-LENS is our CESM-LE). With regard to other aspects of model evaluation, Kay et al. (2015) show that the CESM-LENS simulations compare reasonably well to observational data and other models in the representation of temperature trends and the frequency of atmospheric blocking, with the latter being specifically relevant for heatwaves. Additionally, Jézéquel et al. (2017) studied the exceptionally warm December 2015 comparing sea level pressure data of the CESM-LE to the NCEP Reanalysis 1 (Kalnay et al., 1996) and find no statistically significant differences. However, they also state: "A caveat of this study is that we only used one model, which could have biases especially in the future." This is true for our study, too, and we will also note this in the paper. Additional notes on these evaluation aspects will be added to the manuscript.

**References**

Fischer, E., Sippel, S., and Knutti, R.: Increasing probability of record-shattering climate extremes, Nature Climate Change, 11, 689–695, 2021.

Jézéquel, A., Yiou, P., Radanovics, S., and Vautard, R.: Analysis of the exceptionally warm December 2015 in France using flow analogues, Bulletin of the American Meteorological Society, 99, S76–S79, 2017.

Kalnay, E., Kanamitsu, M., Kistler, R., Collins, W., Deaven, D., Gandin, L., Iredell, M., Saha, S., White, G., Woollen, J., et al.: The NCEP/NCAR 40-year reanalysis project, Bulletin of the American meteorological Society, 77, 437–472, 1996.

Kay, J. E., Deser, C., Phillips, A., Mai, A., Hannay, C., Strand, G., Arblaster, J. M., Bates, S., Danabasoglu, G., Edwards, J., et al.: The Community Earth System Model (CESM) large ensemble project: A community resource for studying climate change in the presence of internal climate variability, Bulletin of the American Meteorological Society, 96, 1333–1349, 2015.

Schaller, N., Sillmann, J., Anstey, J., Fischer, E. M., Grams, C. M., and Russo, S.: Influence of blocking on Northern European and Western Russian heatwaves in large climate model ensembles, Environmental Research Letters, 13, 054 015, https://doi.org/10.1088/1748-9326/aaba55, 2018.

Zschenderlein, P., Fink, A. H., Pfahl, S., and Wernli, H.: Processes determining heat waves across different European climates, Quarterly Journal of the Royal Meteorological Society, 145, 2973–2989, 2019.

---

## Author Response (AR1)

**Responses to the comments of Reviewer 1, 2 and the Editor**

by Lisa Schielicke and Stephan Pfahl

We would like to thank both reviewers and the editor for their helpful comments. This document is structured as follows: We address all comments of reviewers 1 and 2 in section 1 and 2, respectively. In section 3, we provide answers to the editor's comments. Our responses are printed in red and labeled by the abbreviation AC (author comments). The referees and editor's comments are given in black and italic and labeled by RC1, RC2 and EC1, respectively. In addition to responding to the reviewer comments, we have slightly expanded the discussion in section 4 to include a hypothesis on the effect of changes in land-sea temperature contrast.

**1   Comments of Reviewer 1:**

**1.1   Reviewer 1 – General comments:**

RC1: *This manuscript/preprint is devoted to the analysis of European heatwaves under recent past (1991-2000) and future (2091-2100, RCP8.5) climatic conditions. A 3-D Lagrangean description (10-day backward trajectories) of the air mass stream flows underlying the occurrence of percentile-based heatwaves (Heat Wave Magnitude Index daily), with relative and time-dependent thresholds, in gridpoints within six target regions over Europe (following Zschenderlein et al., 2019) is carried out. A 35-member/initialization ensemble, generated by the Community Earth System Model (CESM1), is used for climate change impact assessments.*
*Overall, the topic of research is quite pertinent in the ongoing climate change context, with major and unprecedented heatwaves hitting many regions worldwide. Understanding their driving (dynamical and thermodynamical) mechanisms is of utmost relevance for the scientific community and society in general. The data and methods used are adequate for the study's purposes. State-of-the-art datasets are selected. A sufficient literature review is provided to the readers. The results are scientifically sounding and in agreement with previous research. The text is easy to follow, with enough explanation of the methodologies and findings. The figures are of good quality but too much. Therefore, I recommend the acceptance of this manuscript after some minor revisions that are outlined in the specific comments below.*

AC: Thank you very much for your helpful comments! We have reduced the number of figures and move some of the figures to the supplementary material.

**Reviewer 1 – specific comments:**

RC1: *In the abstract: please state that you have used RCP8.5 in your climate change projections.*

AC: We added a note to the abstract.

RC1: *Section 2.1: Although using a single general circulation model (GCM), i.e. CESM1, is understandable due to the lack of other similar data sources, this is indeed an important limitation of the study that should be stressed. Furthermore, the potential implications of using a single GCM in the results should also be more deeply discussed in Section 4. Only a short sentence is related to this point in the last paragraph of Section 4.*

AC: You are right, we will discuss this limitation in more detail. As we need high-resolution output on model levels for our trajectory calculations, these analyses cannot easily be performed based on other model simulations, e.g., from the CMIP6 archive. Nevertheless, our comparison with the results of Zschenderlein et al. (2019) based on ERA Interim reanalysis data indicate that CESM1 captures the basic dynamics of European heatwaves reasonably well, which gives confidence in the corresponding future projections. We addressed this topic in the discussion section.

RC1: *Section 2.1: "Re-runs of the simulations have been performed for two 10-year time slices...". Please specify who did these new simulations and if they are available for upcoming studies.*

AC: The re-runs were performed by Urs Beyerle (ETH Zurich), as specified in the Acknowledgment section. The data set is very large and can therefore not be easily made publicly available. We would like to encourage

interested readers to contact us in case of questions, and we'd be happy to provide the model output on an individual basis.

RC1: *Section 2.3: The heatwave definition based on the 90th percentile of the daily maximum temperature within a 30-day running window is a reasonable choice. However, something should be mentioned regarding the possible implications of having a higher threshold (e.g., 95th percentile) or a different window length. A sensitivity analysis could be useful to clarify this issue and improve the robustness of the results.*

AC: The heatwave index applied in this study is very well established and has been widely used. The choice of the parameters (90th percentile level, the 5% coverage of land grid points and the $\leq 3$ days) follows Zschenderlein et al. (2019) to allow for a direct comparison with their results based on reanalysis data, which, in our opinion, is an important part of our study. Increasing the percentile threshold and the minimum life time will naturally lead to lower numbers of heatwave cases. Of course it would be interesting to look at the most intense heatwaves and how these will change in the future, which would be an interesting follow-up study. Nevertheless, rerunning the analysis for different thresholds would be a lot of effort (e.g., the present setting requires computing more than 1 million trajectories for the heatwaves), would make an already long paper even longer, and, in our opinion, thus is beyond the scope of this study.

RC1: *Section 2.3: The choice of having different thresholds for each time slice is plausible. Nonetheless, this choice deserves further explanation. In Section 1 this issue is already mentioned, but it would be useful to address this point here as well.*

AC: Thanks! We addressed this topic in more detail. We added the following text to section 2.3: " We decided to use different climatologies for the two time slices since, otherwise, the method would mainly detect heatwaves in the future time slice due to the mean increase in global temperature in the future (see Fig. **??**). As can be seen in Fig. **??**, heatwaves in the historic times slice have mean temperatures comparable to the future mean summer (June-July-August) temperatures. Since the focus of this work is on a comparison of historic and future heatwave characteristics, the heatwave identification needs to be done for each time slice separately. "

RC1: *Line 114: Please explain "annual maximum temperature". Have you also applied the 30-day running window to the heatwave magnitudes? This is important to understand what you mean by "The latter criterion (Md ¿ 0) also makes sure that heatwaves occur during the warmest time of the year, typically the summer months". The heatwave definition deserves a better explanation for a reader not familiar with this concept.*

AC: We added the following, more detailed explanation to the text: "The annual maximum temperature at a grid point represents just one value per year, i.e. the maximum of the 2m temperature observed in this year. Percentiles $T_{\Delta y,25th}$, $T_{,\Delta y,75th}$ are then calculated from the 350 values in each time slice (35 members times 10 years). Since the annual maximum temperature typically occurs in the warm season, $M_d > 0$ is connected to the warmer season."

RC1: *Equation 2: I would say "isobaric advection" rather than "advection", as vertical advection corresponds to the adiabatic expansion/compression term.*

AC: Thanks for your thoughts. However, we disagree in this point: The advection term in eq. 2 represents the full, three-dimensional advection, not just the isobaric one. We add the derivation here for clarification. Note that in the original manuscript there has been a sign error in this equation, which we will correct in the revision:

$$du = c_v dT = -p d\alpha + \delta q \tag{1}$$

$$= -d(p\alpha) + \alpha dp + \delta q \tag{2}$$

$$= -d(RT) + \alpha dp + \delta q \tag{3}$$

$$(c_v + R)dT = \alpha dp + \delta q \qquad\qquad |R = c_p - c_v \qquad . \tag{4}$$

$$c_p dT = \alpha dp + \delta q \tag{5}$$

$$dT = \frac{\alpha}{c_p} dp + \frac{\delta q}{c_p} \qquad\qquad \left| \frac{d}{dt} \right. \tag{6}$$

$$\frac{dT}{dt} = \frac{\alpha}{c_p} \underbrace{\frac{dp}{dt}}_{=\omega} + \frac{1}{c_p} \frac{\delta q}{dt} \tag{7}$$

$$\frac{\partial T}{\partial t} + \vec{v} \cdot \nabla T = \frac{\alpha}{c_p} \omega + \frac{1}{c_p} \frac{\delta q}{dt} \tag{8}$$

where $\alpha = \rho^{-1} = V/m$ is the specific volume; $T$: temperature, $p$: pressure, $R$: gas constant of dry air, $c_v$: specific heat capacity at constant volume, $c_p$: specific heat capacity at constant pressure, $\delta q$: heat, $u$: internal energy, $t$: time.

RC1: *Figs 8-13: I suggest reversing the colour scale in the panels with the differences in parcel density (b-e), as reddish (bluish) colours are more commonly associated with positive (negative) anomalies.*

AC: Thanks for your comment. We replotted and combined the figures indicating the statistical significance (see below) and followed your advice with respect to the color scale.

RC1: *The number of figures is excessive. I suggest using supplementary material for some of them and shortening part of the text, namely sections concerning Figs 2-7. The authors may also choose to present the results only for some of the target regions, leaving the other figures as supplementary material.*

AC: Thanks, you are right. We decided to reduce the number of figures and shift some of them to the supplementary material, e.g. Fig. 4 and 5. We recombined parts of Figs. 8-13 to two figures with focus on the heatwaves and their future changes. The additional information on the comparison to the JJA climatology has been moved to the supplementary material. Moreover, we moved Fig. 19 and 21 to the supplementary material. Overall we now have 13 figures instead of 21.

**Reviewer 1 – Technical corrections:**

RC1: *Lime 114: Delta y instead of Delta t in the 25th percentile.*

AC: Thanks, we changed this.

RC1: *Figure 2: no reference is made to the second box of HWhist.*

AC: Thanks, we added some lines to the caption in order to address all boxes correctly.

RC1: *Line 315: awkward sentence. Please revise.*

AC: We rewrote the sentence: "Regions close to the ocean (BI, GI, IP) have a higher percentage of trajectories in cluster A (see red lines in Fig. XX), which is characterized by diabatic cooling."

RC: *Line 465: "presented her". Please correct.*

AC: We corrected to "presented here".

**2 Reviewer 2 – Comments:**

RC2: *This manuscript addresses an important topic in a novel and informative way. The presentation is clear and the methods appear sound. I have a couple of comments:*
*It is difficult to assess whether the changes projected by the model are significant. Some measure of statistcal significance of the change values is needed for at least some of the analyses. The maps in Figures 8-13 would especially benefit from this*

AC: We performed t-tests to determine the significance of the differences in means (box plots, median trajectories, trajectory properties). Moreover, significance of differences in parcel densities (Figs. 8-13 in original manuscript) was determined by determining 99% confidence intervals at each grid point by randomly drawing 100 times from the original data with replacement (bootstrapping). Grid points marked by a green dot in Figs. 6,7 (revised manuscript) show no significant difference in parcel densities between the respective two distributions.

RC2: *Figures 15-21 could be condensed to a subset of the figures. Part of the challenge is the multiple subregions, which I do see value in including in each analysis. However, I don't think all of the information presented in these figures is necessary as the results tend to show similar conclusions in diferent ways.*

AC: Thanks, you are right. As we wrote above: We reduced the number of figures (from 21 to 13) and shifted some of them to the supplementary material (Figs. 4, 5, 19 and 21 of the original manuscript). Moreover, we recombined parts of Figs. 8-13 to two figures with focus on the heatwaves and their future changes (new Figs. 6 and 7 in revised manuscript). The additional information on the comparison to the JJA climatology has been moved to the supplementary material (Figs. S6, S7). The text has been adapted accordingly.

RC2: *The authors use the 2022 London heatwave to conclude that warming is following the RCP8.5 trajectory. I would suggest mentioning some caveats here. For one, there is growing evidence that emissions are deviating from RCP8.5. Also, using a single event could be a little misleading on assessing a trend. I do think putting this event into context, however, is helpful.*

AC: Thanks, you are right. We have partly rewritten the part concerning the 2022 London heatwave and removed the specific reference to the RCP8.5 scenario, also because up to about 2030 to 2040 all scenarios show similar ranges of temperature change. The main point of this short paragraph is to give an example showing that heatwaves are already changing in the direction as projected for the end of the century by global climate models, which may help readers to get a more concrete idea of such changes to come.
* * *
**3 Editor – Comments:**

EC1: *I would like to thank the two reviewers for their constructive comments, which I hope will be helpful for the authors. I note that both reviewers have suggested that the number of figures could be reduced, so this would be good to attempt in a revised version. I also agree with reviewer 2 that some measure of significance would be beneficial to improve our confidence in the projected changes.*

AC: Thanks, you very much for your comments and also for the reviewers' comments that were very helpful! We reduced the number of figures (from 21 to 13) and shifted some of them to the supplementary material (Figs. 4, 5, 19 and 21 of the original manuscript). Moreover, we recombined parts of Figs. 8-13 to two figures with focus on the heatwaves and their future changes (new Figs. 6 and 7 in revised manuscript). The additional information on the comparison to the JJA climatology has been moved to the supplementary material (Figs. S6, S7). The text has been adapted accordingly. Regarding the significance: We performed t-tests to determine the significance of the differences in means (box plots, median trajectories, trajectory properties). Moreover, significance of differences in parcel densities (Figs. 8-13 in original manuscript) was determined by determining 99% confidence intervals at each grid point by randomly drawing 100 times from the original data with replacement (bootstrapping). Grid points marked by a green dot in Figs. 6,7 (revised manuscript) show no significant difference in parcel densities between the respective two distributions.

EC1: *I also have one additional comment. Although this is a widely-used model it would still be good to touch on the issue of model evaluation, since all of the results rely on the fidelity of this model. How similar are the detected events in this model to those in observations and/or reanalyses, and how does the model perform in evaluation / validation exercises in aspects relevant to European heatwaves? Hopefully this could be addressed using existing literature rather than new data analysis.*

AC: Our methodology closely follows the study of Zschenderlein et al. (2019), who performed a similar analysis based on ERA-interim reanalysis data. This has enabled us to compare our results on heatwaves in the historical time slices with theirs, which has already been described at several places in the original manuscript. As discussed there, there is a reasonable agreement with regard to general heatwave statistics (see section 3.1 of the original manuscript), composite patterns (section 3.2), and air parcel origins (section 3.3). For the latter, Zschenderlein et al. (2019) found the maximum of parcel density three days prior to the heat event onset mainly inside of the regions of interest, while in CESM it typically is slightly displaced, but still in close proximity. Nevertheless, the overall agreement makes us confident that we can also trust the future projections of European heatwave dynamics of CESM. This is emphasized more explicitly also in the conclusions section of the revised manuscript. Furthermore, Schaller et al. (2018) show that the relationship between European heatwaves and blocking (which is central for the circulation associated with heatwaves) is properly represented in CESM-LENS, also in comparison to another climate model ensemble. Nevertheless, CESM is not able to reproduce events as extreme as the Russian heatwave in 2010, which may point to model biases in the representation of very extreme events. At the same time, Fischer et al. (2021) find, in a gridpoint based analysis of warm extremes, that the probability for "record-shattering" events, i.e., events larger or equal to twice the standard deviation, for the large ensembles of the CESM-model family is at the higher end of the model range compared to CMIP5 and CMIP6 ensembles (note: their ensemble NCAR-LENS is our CESM-LE). With regard to other aspects of model evaluation, Kay et al. (2015) show that the CESM-LENS simulations compare reasonably well to observational data and other models in the representation of temperature trends and the frequency of atmospheric blocking, with the latter being specifically relevant for heatwaves. Additionally, Jézéquel et al. (2017) studied the exceptionally warm December 2015 comparing sea level pressure data of the CESM-LE to the NCEP Reanalysis 1 (Kalnay et al., 1996) and find no statistically significant differences. However, they also state: "A caveat of this study is that we only used one model, which could have biases especially in the future." This is true for our study, too, and is also noted in the revised manuscript. Additional notes on these evaluation aspects have been added to the manuscript.

**References**

Fischer, E., Sippel, S., and Knutti, R.: Increasing probability of record-shattering climate extremes, Nature Climate Change, 11, 689–695, 2021.

Jézéquel, A., Yiou, P., Radanovics, S., and Vautard, R.: Analysis of the exceptionally warm December 2015 in France using flow analogues, Bulletin of the American Meteorological Society, 99, S76–S79, 2017.

Kalnay, E., Kanamitsu, M., Kistler, R., Collins, W., Deaven, D., Gandin, L., Iredell, M., Saha, S., White, G., Woollen, J., et al.: The NCEP/NCAR 40-year reanalysis project, Bulletin of the American meteorological Society, 77, 437–472, 1996.

Kay, J. E., Deser, C., Phillips, A., Mai, A., Hannay, C., Strand, G., Arblaster, J. M., Bates, S., Danabasoglu, G., Edwards, J., et al.: The Community Earth System Model (CESM) large ensemble project: A community resource for studying climate change in the presence of internal climate variability, Bulletin of the American Meteorological Society, 96, 1333–1349, 2015.

Schaller, N., Sillmann, J., Anstey, J., Fischer, E. M., Grams, C. M., and Russo, S.: Influence of blocking on Northern European and Western Russian heatwaves in large climate model ensembles, Environmental Research Letters, 13, 054 015, https://doi.org/10.1088/1748-9326/aaba55, 2018.

Zschenderlein, P., Fink, A. H., Pfahl, S., and Wernli, H.: Processes determining heat waves across different European climates, Quarterly Journal of the Royal Meteorological Society, 145, 2973–2989, 2019.